# SELF-GENOMENET: SELF-SUPERVISED LEARNING WITH REVERSE-COMPLEMENT CONTEXT PREDICTION FOR NUCLEOTIDE-LEVEL GENOMICS DATA

## ABSTRACT

In this paper, we introduce a novel contrastive self-supervised method, namely Self-GenomeNet, for nucleotide genome representation learning. To the best of our knowledge, Self-GenomeNet is the first self-supervised framework that learns a representation of nucleotide genomic data, using domain-specific characteristics. Our proposed method learns and parameterizes the latent space by leveraging the reverse-complement of genomic sequences. During the training procedure, we force our framework to capture semantic representations with a novel context network on top of features extracted by an encoder network. The network is trained with an unsupervised contrastive loss. Extensive experiments with different datasets show that our method with self-supervised and semi-supervised settings outperforms state-of-the-art deep learning methods. Furthermore, we show that the learned representations generalize well and can be transferred to new datasets and tasks. The source code of Self-GenomeNet and all experiments are provided as supplementary material.

## 1 INTRODUCTION

In bioinformatics, learning from unlabeled data can reduce development costs in many deep learning applications, since it reduces the number of required annotated samples that often require manual experimental validation, such as the functional annotation of genes (Gligorijević et al., 2021) or chromatin effects of single nucleotide polymorphisms (SNPs) (Zhou and Troyanskaya, 2015). Self-supervised learning is amongst the most promising approaches for learning from limited amounts of labeled data and is an attractive approach in bioinformatics due to the availability of large quantities of unlabeled sequence data. In contrast to supervised methods, these techniques learn a representation of the data without relying on human annotation. Early self-supervised methods focused on solving a *pretext task* (Devlin et al., 2018) that allows utilizing the data itself to generate labels, and use supervised methods to solve unsupervised problems. The representations learned by performing this task can be used as a starting point for *downstream* supervised tasks such as taxonomic prediction or gene annotation. Several different self-supervised representation learning methods were proposed e.g. in the fields of natural language processing (NLP) and computer vision (CV). However, self-supervised learning has not yet seen such widespread adoption in bioinformatics and remains an important and challenging endeavor for the representation of genome data.

Existing methods for representation learning for omics-data tends to use methods from NLP or CV with slight architecture adjustments and additional preprocessing (e.g. Rives et al., 2021; Lu et al., 2020; Ciortan and Defrance, 2021). However, these methods so far do not take many biological characteristics into account. In particular, the genome sequence embodied by one strand on the DNA molecule is always accompanied by another strand going in the opposite direction with complementary nucleotides. This means that both the forward and the so-called *reverse-complement* (RC) sequences are valid DNA sequences and equally correspond to several properties of the genome, such as regulatory and taxonomical characteristics. Therefore, machine learning models developed to distinguish different aspects of genomic sequences ideally take the RC into account, for example by being equivariant with respect to it (Mallet and Vert, 2021; Zhou et al., 2021). Our proposed Self-GenomeNet uses a context network that learns representations that also take advantage of the RC.

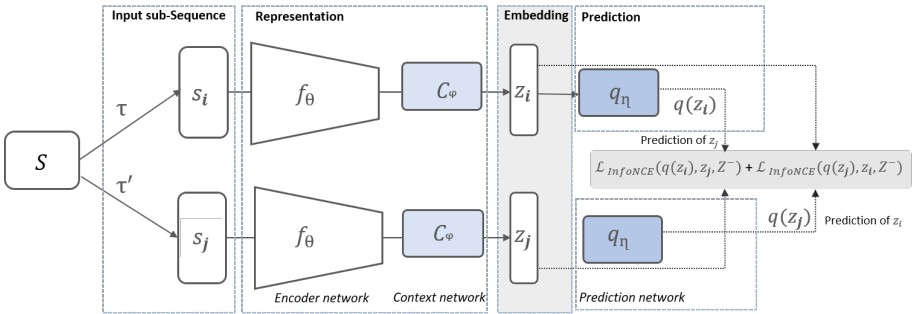

Figure 1: Illustration of our proposed Self-GenomeNet. Conceptually, our method splits a sequence $S$ into two subsequences $s_i, s_j$ and uses the reverse-complement $\overline{s_j}$ of the second subsequence. The subsequences are fed into a representation block composed of the encoder network $f_\theta$, followed by the context network $C_\phi$ to produce the embeddings $z_i, \overline{z_j}$. Each of the two embeddings predicts the other one, against embeddings of other sequences in the same batch, using an unsupervised contrastive loss.

We propose a simple contrastive self-supervised algorithm for learning representations of nucleotide-level genomic sequences (Figure 1). Our method divides a sequence into two non-intersecting subsequences, where one of them is transformed to its RC. Then, the contextual representations of these subsequences are required to distinguish each other from the representations of other, random sequences. We exploit the inherent symmetry in the genome sequence implied by the RC to achieve a natural simplification over possible methods that do not take it into account, all while encoding RC-awareness in our architecture. Moreover, we train our network with varying-length subsequences both for the input and the target subsequence. Because, genome sequences can have semantic structure at varying ranges of length, representations that respect both local and global structures are preferable. For this, the context network makes use of a recurrent neural network layer, which makes the computationally efficient training based on all used length scales within a single step possible. Our self-supervised method utilizes contrastive loss to maximize the similarity between representations of subsequence and its RC while minimizing the similarity between representations derived from unrelated sequences.

**Contributions:** We present the first self-supervised framework tailored specifically to DNA sequence data. Our main findings and contributions can be summarized as follows: (1) We introduce a contrastive self-supervised method that is specialized for nucleotide-level genomics data. (2) We propose a context-aware network that makes use of inherent symmetries in genomics data in the form of the reverse-complement and improves the quality of learning representation. (3) Self-GenomeNet is evaluated in self-supervised and semi-supervised settings on bioinformatic benchmarks and datasets and achieved superior performance in several tasks. (4) We show the learned representations generalize well when transferred to a new dataset and task.

## 2 RELATED WORK AND BACKGROUND

**Self-supervised learning** Deep learning models based on self-supervised learning have become a standard method of choice in natural language processing (NLP). Powerful models such as GPT-3 (Brown et al., 2020) and BERT (Devlin et al., 2018) are pre-training in a self-supervised manner on a large text corpus. They are then often fine-tuned for various tasks using smaller task-specific datasets, where they show performance that is superior to purely supervised methods. Recently, self-supervised visual representation learning (Chen et al., 2020) outperformed supervised methods and achieved state-of-the-art performance on ImageNet (Deng et al., 2009). Self-supervised contrastive learning is one of the most successful methods where it maximizes the agreement of learned representations between differently augmented views of the same example (Chen et al., 2020; Oord et al., 2018). Motivated by this, we propose a novel contrastive self-supervised model similar to (Oord et al., 2018) but differing in network architecture, representation learning strategy, and with applications specifically in bioinformatics.

**Self-supervised learning in bioinformatics** Existing self-supervised learning techniques applied on sequential RNA and protein data are adopted from other application fields of deep learning. (Rives

et al., 2021) adopted BERT (Devlin et al., 2018) as a pre-trained model for predicting masked amino acids and achieved superior performance on sequence prediction task. Contrastive-sc (Ciortan and Defrance, 2021) is a method adapted from computer vision for cell clustering based on single-cell RNA sequencing data. Contrastive-sc creates two copies of sequences with randomly masked nucleotides and trained the network using a contrastive loss. As a downstream task, self-supervised embeddings are used for clustering with K-Means algorithm (Hartigan and Wong, 1979). All these methods for self-supervision described so far have in common that they rely on randomly masking part of their input to create a pretext task. This differs from Contrastive Predictive Coding (CPC) (Oord et al., 2018), which splits the input into parts such as image patches or subsequences, and predicts the representation of some of the parts, contrastively, from the representation of other parts. CPCProt (Lu et al., 2020) adapted the CPC model for protein data.Similarly, Self-GenomeNet is trained by the contrastive loss in which the positive pairs include the input sequence and a reverse-complement of the sequence. Unlike previous works, our network is trained with varying-length targets to explore both local and global structures of sequences.

**Deep learning methods for genome sequences** In genomics, many deep learning architectures use convolutional layers to capture local patterns in data, whereas recurrent layers learn long-range dependencies in the sequences (Eraslan et al., 2019). Zhou and Troyanskaya (2015) created the DeepSEA dataset for predicting chromatin effects of sequence alterations, and used it to train a model consisting of multiple convolutional layers. Later, DanQ (Quang and Xie, 2016) improved the deep learning framework of DeepSEA using a bidirectional LSTM (Hochreiter and Schmidhuber, 1997) unit on top of a convolutional layer. iDeepS (Pan et al., 2018), DeepBind (Alipanahi et al., 2015), and DeeperBind (Hassanzadeh and Wang, 2016) use a sequence CNNs and RNNs to predict RNA binding proteins sites. DeepDNA (Wang et al., 2019) uses an LSTM layer on top of a CNN layer to compress human mitochondrial genome data. Instead of recurrent layers, it is also possible to use dilated convolutional layers to model long-range dependencies in genome data, such as done in Basenji (Kelley et al., 2018), which is used to predict cell-type–specific epigenetic and transcriptional profiles in large mammalian genomes. Trabelsi et al. (2019) performed a comprehensive study on network architecture where the ECBLSTM network, composed of an embedding layer, CNNs, and LSTM layer outperformed other networks on several genomic tasks. Here, we also used the DeepSEA dataset to compared the performance of our method with several other self-supervision techniques. The architecture of Self-GenomeNet requires a recurrent layer for it to create representations of varying-length subsequences, and is therefore based off the DanQ-architecture (Quang and Xie, 2016), which has shown good performance on the DeepSEA task.

**Reverse Complement** DNA molecules in living organisms always occur as a double strand, where both strands have opposite directions and consist of nucleotides that are complementary to each other. The sequence on one strand is therefore accompanied by its *reverse complement* (RC) on the other: The sequence in reverse, and with the *A, C, G, T* nucleotides replaced by *T, G, C, A* respectively. Since both forward and reverse-complement sequences are found in the structure, they are equally responsible for several properties of the genome, such as regulatory and taxonomical characteristics. However, next-generation sequencing techniques (NGS) output sub-sequences randomly drawn from both strands, since it usually involves breaking the genome into a collection of small DNA fragments that are sequenced individually. Therefore, both the original sequence data and the RC of this sequence are valid inputs to the machine learning models developed to work on NGS datasets (Shrikumar et al., 2017; Zhou et al., 2021). In this paper, we exploit this genome-specific feature in contrastive self-supervised learning framework helps to improve the learning representation in several genomic tasks.

## 3 METHOD

We first describe the motivation of our proposed method before explaining its details in Section 3.2.

### 3.1 MOTIVATION

Most successful self-supervised learning approaches are built for computer vision or NLP and are therefore not optimal for bioinformatics data. For example, the self-supervision task of CPC (Oord et al., 2018) or generative language models (Dai and Le, 2015) are based on predicting the representation of a short, constant length subsequence or a single token, which could be an N-gram or a single nucleotide. We hypothesize single-length short sequence targets hinder the model from learn-

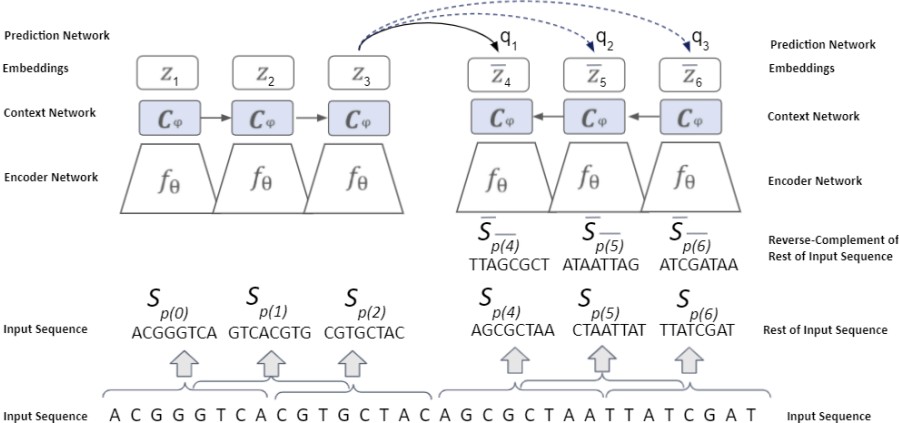

Figure 2: Implementation of Self-GenomeNet. Multiple representations $\mathbf{z}_i$ of subsequences of different lengths are created in a single step using the sequence-output of the recurrent context network $C_\phi(\cdot)$ which follows the encoder network $f_\theta(\cdot)$. The same process happens to the reverse-complement of the given sequence, creating $\overline{\mathbf{z}}_i$. Different prediction layers $q_m$ are used to predict representations that are a different number of patches apart.

ing global semantics effectively. We therefore propose training representations that are based on target subsequences whose lengths are varying. While predicting future observations of constant sequence length from input sequences which are changing in length is common for methods like CPC and single character models, predicting representations of varying length sequences is a novelty in this area. Similar to Oord et al. (2018), we use a recurrent context network on top of an encoding network that encodes to constant-size patches. However, we use the output of the context network to predict the output of the same context network applied to other patches, as opposed to predicting only the output of the encoding network. The fact that the RC of an input sequence is a valid genome sequence as well simplifies this approach: instead of having to fit both one forward and one reverse context network that predict each other, only a single context-network is trained that is both applied to the forward as well as to the RC sequence.

### 3.2 SELF-GENOMENET

The Self-GenomeNet architecture learns representations of varying-length genome sequences by encoding them in both convolutional and recurrent neural network layers, which are trained using a contrastive loss. The architecture enforces the symmetry implied by the RC while also making training more efficient. As depicted in Fig. 1, our proposed method initially creates two subsequences from an input sequence. The subsequences are taken by a representation network composed of the *encoder network* ($f_\theta$) and the *context network* ($C_\phi$). Later, on top of the embedding representation, the linear prediction layer ($q_m$) estimates the embedding of the other subsequence using a contrastive loss against random other subsequences. More specifically, given are an unlabeled genome sequence of nucleotides $S_{1:N} = [s_1, s_2, \ldots, s_N]$ and its RC $\overline{S}_{N:1} = [\overline{s_N}, \overline{s_{N-1}}, \ldots, \overline{s_1}]$, where $s_i \in \{A, C, G, T\}$ and $\overline{s_i}$ gives the complementary nucleotide, e.g. $\overline{A} = T$. We aim to produce an embedding of a subsequence $S_{1:t}$ that contrastively predicts the embedding of the RC of the remaining subsequence, $\overline{S}_{N:t'}$ for some $t' > t$. Predicting $\overline{S}_{N:t'}$, i.e. the RC of $S_{t':N}$, instead of just the reverse of $S_{t':N}$, has the benefit that it is possible to use the same embedding and context network for both the first and the second subsequence, as $\overline{S}_{N:t'}$ is a valid genome sequence, whereas $S_{N:t'}$ is not. We use a recurrent layer to build the representations of both $S$ and $\overline{S}$, so representations of different length subsequences are calculated at once by using the hidden state of the recurrent unit at different points in the sequence. This makes it possible to contrastively predict subsequences of varying lengths against each other efficiently in a single training step. This has an advantage over using two forward sequences $S_{1:t}$ and $S_{t':N}$, where using different starting point values $t'$ in one step would be much more computationally demanding, since the values of recurrent units would have to be calculated fresh for every $t'$. The entire procedure is symmetric since our model learns sequence and the RC of sequence. This symmetry provides the possibility to use a context network as well

as the prediction networks with shared weights both when reading the subsequence in forward direction as in the left subsequence in Fig. 2 as well as when reading the reverse-complement of the sequence on the right. This is visualized in Fig. 3 in Appendix. (i) The same recurrent context network is used for predicting and target subsequences, (ii) the target subsequence can also predict the predicting subsequence, (iii) the same prediction network(s) used for predictions of the target subsequences from the predicting subsequences can also be used for the predictions of the predicting subsequences from the target subsequences. In other words, with the help of the symmetry induced by RC, substantial weight sharing is provided.

To produce the representation, the sequence is first divided into $P$ overlapping patches $S_{p(j)}$ with $p(j) = (j \cdot a + 1){:}(j \cdot a + l)$, where $l$ is the patch-length and $a$ the patch stride length, and where $a < l$ for overlapping patches. Patches are first encoded with a convolutional neural network $f_\theta(\cdot)$. The resulting sequence of vectors $[f_\theta(S_{p(0)}), f_\theta(S_{p(1)}), \ldots, f_\theta(S_{p(P-1)})]$ is fed into a recurrent context network $C_\phi(\cdot)$, giving rise to embeddings $\mathbf{z}_i = C_\phi(\{f_\theta(S_{p(u)})\}_{u<i})$ for $1 \leq i \leq P$. Patches of the RC, $\overline{S}_{\overline{p}(j)} = [\overline{s_{j \cdot a + l}}, \ldots, \overline{s_{j \cdot a + 1}}]$ are equally fed into first $f_\theta$ then $C_\phi$, giving rise to $\overline{\mathbf{z}_i} = C_\phi(\{f_\theta(\overline{S}_{\overline{p}(u)})\}_{u \geq i})$. $\mathbf{z}_i$ is then a representation of $S_{1:((i-1)\cdot a + l)}$; likewise, $\overline{\mathbf{z}}_i$ represents $\overline{S}_{N:(i \cdot a + 1)}$.

Training the encoder and context networks consists of predicting $\overline{\mathbf{z}}_{i+m}$ from $\mathbf{z}_i$, with $m > 0$, contrastively against corresponding embeddings from other, negative example sequences $S^{(k)-}$, i.e. against $\overline{\mathbf{z}}_{i+m}^{(k)-}$ (note: $m$ is the stride length between the patches). For this, a linear prediction layer $q_m$ is used, and the Noise Contrastive Estimation or InfoNCE loss (Gutmann and Hyvärinen, 2010), which maximizes the mutual information shared between forward sequence and its matching RC sequence, is used:

$$L_{i,m} = -\log \frac{\exp\left(\overline{\mathbf{z}}_{i+m}^T q_m(\mathbf{z}_i)\right)}{\exp\left(\overline{\mathbf{z}}_{i+m}^T q_m(\mathbf{z}_i)\right) + \sum_k \exp\left(\left(\overline{\mathbf{z}}_{i+m}^{(k)-}\right)^T q_m(\mathbf{z}_i)\right)}. \tag{1}$$

Negative samples $\overline{\mathbf{z}}_{i+m}^{(k)-}$ are efficiently produced by comparing against representations of other sequences loaded in the same mini-batch. Note that every sequence in the minibatch produces *two* negative examples for other sequences, giving $2(B-1)$ negative samples when using minibatch size $B$: The embedding of the RC of the last $(P - i - m)$ patches starting from path $(i + m)$ – i.e., the analog to $\overline{\mathbf{z}}_{i+m}$ coming from other sequences – and the embedding of the first $(P - i - m)$ patches of the normal forward-sequence, so the analog to $\mathbf{z}_{P-i-m}$. This is because of the invariance of the training method with respect to the RC. Embeddings are, however, always only contrasted with embeddings of sequences that have the same length as the positive example. This is to prevent the network from learning to encode the represented length directly to gain an advantage, which would not be an intrinsically interesting feature for downstream tasks.

One loss term is introduced for each index $i$, denoting the number of patches represented by $\mathbf{z}_i$. Furthermore, because it is possible to skip a different number of patches $m$ between the representations being matched, one can sum over multiple values of $m$, each with its individual weights for predictor $q_m$. Because of the symmetry of the setup, the model is both predicting $\overline{\mathbf{z}}_{i+m}$ from $\mathbf{z}_i$ as described above, as well as $\mathbf{z}_i$ from $\overline{\mathbf{z}}_{i+m}$. The corresponding loss $\overline{L}_{i,m}$ then uses negative examples with the same length as $\mathbf{z}_i$. The ultimate loss for each individual sequence $S$ is therefore:

$$L = \sum_{i,m} \left(L_{i,m} + \overline{L}_{i,m}\right). \tag{2}$$

It was proven by (Oord et al., 2018), optimizing a contrastive loss by predicting future patches $S_p(j)$ maximizes the lower bound of mutual information between the patches and the embeddings. The theoretical proof regarding how we maximize the lower bound of mutual information between the embeddings $z_i$ and varying-length reverse-complement targets $\{\overline{S}_{\overline{p}(u)}\}_{u \geq j}$ for several values of $j$ is straight-forward by exchanging $\{\overline{S}_{\overline{p}(u)}\}_{u \geq j}$ and $S_p(j)$ (they use $x_{k+t}$ as notation) in the equations of the proof derived by Oord et al. (2018). By adopting the equations accordingly, we derive,

$$I(\{\overline{S}_{\overline{p}(u)}\}_{u \geq j}, z_i) \geq log(N) - L \tag{3}$$

where $I(\{\overline{S}_{\overline{p}(u)}\}_{u \geq j}, z_i)$ is the mutual information between the learned representation $z_i$ and the reverse-complement of the consecutive sequence $\{\overline{S}_{\overline{p}(u)}\}_{u \geq j}$. $N$ is the number of random samples in the contrastive pretraining and $L$ is the loss function given in the Eq.2. Note that in our method, differently from CPC (Oord et al., 2018), as the value of $j$ changes, the number of patches in the set and therefore the total length of the targeted patches vary. By doing this, we maximize the mutual information between varying-length targets and the learned representations.

## 4 EXPERIMENT SETUP

**Datasets and Tasks** We evaluate our method with the following datasets:

(1) **DeepSEA** dataset: introduced by Zhou and Troyanskaya (2015), contains around 5 million human genome sequences, where each data sample contains 1000 nucleotides as input and a label vector for 919 binary chromatin features. We evaluated our model with the DeepSEA dataset because it is a dataset often used by genomic deep learning methods for evaluation (e.g. Zhou and Troyanskaya, 2015; Quang and Xie, 2016; Kelley et al., 2018). It represents the task of identifying the class of certain *regions* within a genome sequence.

(2) **Virus** dataset: We downloaded all publicly available viral genomes from GenBank (Sayers et al., 2021). We allocated the FASTA files into training, validation, and test sets by approximate ratios of 70%, 20%, and 10%. We divided the dataset into two taxonomical classes as a classification task: bacteriophages vs. viruses that are not bacteriophages, based on provided annotations. The bacteriophage class contained approximately 1.0 billion nucleotides, the non-phage virus dataset approximately 0.5 billion nucleotides. The *Virus* dataset is a machine learning task that is complementary to it: instead of regions of a genome, the class of the whole sequence should be determined, based on a given subsequence of fixed length. Species classification tasks often arise in metagenomics, where NGS is used to analyse DNA found in environmental samples (Pust and Tümmler, 2021). NGS typically produces 150nt samples (Quail et al., 2012), therefore a sequence length of 150nt is used with this dataset, which is drawn from the sequences at random. The length of 150nt would therefore allow a direct application of our model in metagenomics.

(3) **Bacteria** dataset: We downloaded all publicly available bacteria genomes from GenBank, comprising approximately 83 billion nucleotides, similarly to how we collected the *Virus* dataset. This dataset was only used for unsupervised pre-training for transfer learning, so no supervised task was defined. The sequence length was set to 1000 nt, to compare the capabilities of the methods for different sequence lengths.

(4) **T6SS** effector protein dataset: We performed experiments on *T6SS* dataset that publicly available bacteria dataset (SecReT6, Li et al. 2015) to demonstrate that our method works well for a real-world dataset with actual label scarcity. Here the task is the identification of effector proteins. We used T6SS effector proteins as the positive samples to identify, whereas T6SS immunity proteins, T6SS regulators, and T6SS accessory proteins are negative samples. We divided the training, validation, and test set by approximate ratios of 60%, 20%, and 20%. The sequence length is 1000nt in all experiments. We provide more details for pre-processing and motivation of the datasets in Appendix B.

**Network architecture** As mentioned in Section 3, Self-GenomeNet composed of a representation block followed by an embedding layer and a prediction network. The representation block includes encoder network $f_\theta(\cdot)$ and context network $C_\phi(\cdot)$. For the self-supervised training on the *Virus* dataset, the encoder network is a convolution layer with 1024 filters, a kernel size of 24, and a stride of 6. Only a single CNN-layer is used, so the kernel parameters directly correspond to patch parameters described above, i.e. $l = 24$ and $a = 6$. The context network is an LSTM layer with a hidden layer size of 512. For the *DeepSEA* benchmark, we followed the DanQ network architecture by Quang and Xie (2016). The encoder network has an additional max-pooling layer followed by a dropout layer.

For each value of $m$, a different fully connected layer is used as the prediction layer $q_m$. In any self-supervised pretraining, the default choice is to have only one prediction network $q_1$ due to longer pretraining time when multiple prediction layers are used. Therefore the default is estimating the neighbor subsequence of the predicting subsequence without any skipped nucleotide between them. However, we additionally test pretraining with multiple prediction networks and therefore estimate

even further subsequences for the *DeepSEA* dataset. The reported results are the best performance of Self-GenomeNet. We provide more information in Appendix A.

**Optimization** We used Adam optimizer (Kingma and Ba, 2014) with $\beta_1 = 0.9$, $\beta_2 = 0.999$ and a learning rate of 0.0001 for all experiments except the DeepSEA dataset that we used the RMSprop optimizer. The dropout rate of 0.2 and 0.5 are used respectively for the dropout layers as in the DanQ model (Quang and Xie, 2016). For all datasets, we used the default Keras weight initialization, which is the Glorot uniform initialization (Glorot and Bengio, 2010). During the self-supervised pre-training, we used the largest fitting minibatch size (power of 2) that worked in a GeForce RTX 2080 Ti, which was 512 for the virus dataset and 128 for the *Bacteria* and *DeepSEA* datasets.

For the baselines experiments, we used the same hyperparameter settings for all our experiments such as Adam hyperparameters or learning rate. When not applicable, we followed the recommendations in the papers of the baseline methods regarding the hyperparameter selection. We provided more detail on optimization in Appendix B.

**Evaluation** For all experiments, we report the performance in terms of $F_1$ score. However, We use Receiver Operating Characteristics (ROC) Area Under Curve (AUC) and Precision-Recall (PR) AUC for *DeepSEA* dataset for all 919 binary outputs, as these metrics are used by Quang and Xie (2016). In this paper, it was also shown that the sparsity of positive binary targets($\sim 2\%$) in the dataset inflated the ROC AUC, and that PR AUC is a better indicator of performance since it does not take the number of true negatives into account. We macro-averaged recall $Recall_M$ (Sokolova and Lapalme (2009)), which is equivalent to balanced accuracy, and $F_1$ score for dataset evaluations on *Virus*.

## 5 RESULTS

We compare our proposed Self-GenomeNet with CPC (Oord et al., 2018) as the CPC is the closest self-supervision method to our method which can also be applied in different application fields of deep learning. Moreover, we provide comparison with Contrastive-sc (Ciortan and Defrance, 2021) and generative language model since these methods originally proposed for computer vision or natural language processing respectively but applied in bioinformatics several times. Both single nucleotides, denoted as (single nt), and 3-gram targets are predicted for pretraining using a language model (See Fig. 4).We follow the standard evaluation protocol for self-supervised learning and evaluate the unsupervised learned representation with a linear classification and semi-supervised tasks, as well as using transfer learning to different datasets and different tasks (Henaff, 2020).

**Linear Evaluation** The common evaluation protocol for self-supervised learning is based on freezing the representation networks (both base-encoder and context network) after unsupervised pre-training and then train a supervised linear classifier on top of them. The linear classifier is a fully connected layer followed by softmax-classification using cross-entropy loss, which is plugged on top of the context network after removing the prediction layer. Table 1 shows the comparison of our model against the baselines under the linear evaluation on Virus and DeepSEA dataset.

| Dataset | DeepSEA | | Virus | |
|---|---|---|---|---|
| Metric | *ROC AUC* | *PR AUC* | $Recall_M$ | $F_1$ |
| CPC | 0.734 | 0.094 | 58.7 | 0.607 |
| Language Model (single nt) | 0.699 | 0.069 | 58.8 | 0.598 |
| Language Model (3-gram) | - | - | 61.8 | 0.626 |
| Contrastive-sc | 0.682 | 0.056 | 54.9 | 0.462 |
| Self-GenomeNet | **0.757** | **0.120** | **69.1** | **0.709** |

Table 1: Mean values of ROC AUC and PR AUC metrics for the 919 outputs defined for the *DeepSEA* dataset and Macro-averaged recall (in %) and $F_1$ for *Virus* under linear evaluation.

**Semi-supervised Learning** We evaluate the performance of our models on a semi-supervised learning task. In this task, we pretrain our model on unlabeled datasets and fine-tune a classification or prediction model using a subset of datasets with labels. Differently from the linear evaluation protocol, the pretrained base network is also fine-tuned. We follow the semi-supervised protocol of (Henaff, 2020) and use the same fixed splits of respectively 1% and 10% of labeled training data in DeepSEA and Virus datasets.

| Dataset | DeepSEA | | | | Virus | | | |
| Labeled | 1% | | 10% | | 1% | | 10% | |
| Metric | $ROC$ | $PR$ | $ROC$ | $PR$ | $Recall_M$ | $F_1$ | $Recall_M$ | $F_1$ |
|---|---|---|---|---|---|---|---|---|
| Supervised | 0.776 | 0.119 | **0.865** | **0.239** | 69.3 | 0.694 | 72.4 | 0.732 |
| CPC | 0.780 | 0.125 | 0.863 | 0.236 | 69.4 | 0.700 | 74.4 | 0.753 |
| Language Model (single nt) | 0.774 | 0.116 | 0.859 | 0.232 | 71.5 | 0.727 | 75.5 | 0.760 |
| Language Model (3-gram) | - | - | - | - | 72.0 | 0.732 | 76.3 | 0.766 |
| Contrastive-sc | 0.764 | 0.105 | 0.848 | 0.222 | 70.3 | 0.706 | 73.2 | 0.742 |
| Self-GenomeNet | **0.795** | **0.138** | **0.865** | 0.235 | **75.3** | **0.751** | **78.2** | **0.785** |

Table 2: Performance of the self-supervision techniques under semi-supervised learning protocol using 1% and 10% of the labels in *DeepSEA* and *Virus* datasets. Supervised training performance without any pretraining is also reported as a baseline. Reported metrics are in line with Table 1.

**Transfer to Other Tasks** We further assess the generalization capacity of the learned representation on viral detection by pre-training Self-GenomeNet on the *Bacteria* datasets. Our motivation for transfer learning is to take advantage of abundant bacterial data, which are much better described and studied than their viral counterparts, and optimize the model for tasks on viruses(such as viral identification or taxonomic classifications). We also used the *T6SS* dataset to demonstrate that our method works well for a real-world dataset with actual label scarcity. To this end, the Self-GenomeNet is trained in self-supervised fashion on the bacteria dataset and then we train in a supervised manner using the phage and non-phage labels on the virus dataset and evaluate the performance on this dataset. Table 3 provides a comparison of transfer learning performance of our self-supervised approach for the task of classification of viruses.

| Dataset | Virus | | | | T6SS | | | |
| Base Network | Fine-tuned | | Fixed | | Fine-tuned | | Fixed | |
| Metric | $Recall_M$ | $F_1$ | $Recall_M$ | $F_1$ | $Recall_M$ | $F_1$ | $Recall_M$ | $F_1$ |
|---|---|---|---|---|---|---|---|---|
| Supervised | 92.0 | 0.938 | - | - | 58.2 | 0.624 | - | - |
| CPC | 95.5 | 0.966 | 76.1 | 0.812 | 81.8 | 0.855 | 72.3 | 0.725 |
| LM (single nt) | 92.6 | 0.944 | 67.9 | 0.735 | 79.1 | 0.825 | 72.5 | 0.752 |
| Contrastive-sc | 93.4 | 0.950 | 56.4 | 0.578 | 74.3 | 0.775 | 62.0 | 0.627 |
| Self-GenomeNet | **96.7** | **0.974** | **82.1** | **0.861** | **83.4** | **0.857** | **79.3** | **0.796** |

Table 3: The achieved performance terms of Macro-averaged recall (in %) and $F_1$ score under transfer learning evaluation on virus dataset (Column 2-5) and T6SS (Column 6-9). We evaluated the results when base networks are fine-tuned and fixed. Here, the base network is trained in self-supervised way on the big bacteria dataset. Supervised training performance without any pretraining is also reported as a baseline.

# 6 ABLATION STUDIES AND DISCUSSIONS

We present multiple ablation studies on Self-GenomeNet to give an intuition of its behavior and performance: (1) the impact of reverse-complement targets against other potential targets; (2) predicting varying-length targets against single-length targets; (3) robustness of our algorithm in the low-data regime. All the ablation study is performed on the virus dataset under linear evaluation setting.

**Learning from the reverse complement of sequences** Self-GenomeNet uses the embedding representation of a context network for downstream tasks which learns by predicting the reverse complement of neighbor subsequence $\bar{S}_{N:t'}$ as a target for its predictions. We investigated the effect of using target embeddings derived by RC of neighbor subsequences against two other potential ways of creating embeddings derived from varying-length targets. To this end, besides our method, we performed two other experiments where the recurrent context network reads the "target" subsequences in different ways: in *forward* and *reverse* directions. In *forward* setting, the context network reads the target subsequence $S_{t':N}$ in the same direction as it reads the predicting subsequence $S_{1:t}$; whereas *reverse* direction refers to reading the target sequence other way around, as in $S_{N:t'}$. Table 4 compares the performance of our trained network in these ways.

| Forward | Reverse | Reverse-Complement |
|---|---|---|
| 0.7048±0.0025 | 0.6952±0.0003 | 0.7089± 0.0008 |

Table 4: The achieved performance in terms of $F_1$ score for virus dataset under linear evaluation setting with fixed pre-trained weights.

**Predicting varying-length targets** We investigate the effect of having varying-length target subsequences by comparing our method against the constant-length target subsequences. As mentioned, having non-varying length targets is indeed a common feature for self-supervision techniques such as CPC or language models. Nevertheless, we would like to investigate the effect of varying-length targets, when other features of Self-GenomeNet is unchanged. To create nonvarying-length target subsequences we divide the sequences into two nearly equal-in-size subsequences. Specifically, we split the sequences of 150 into 72 and 78, so that subsequences with length 72 and 78 predicts the embeddings derived from the reverse-complement of each other. In the Self-GenomeNet however, for the given architecture, subsequences with length $x \in \{24, 30, 36, ...126\}$ predicts a subsequence with length $150 - x$. 72 and 78 was chosen because they are the two closest values that can be investigated using our previous architecture without any padding (splitting into 2 subsequence with 75 is not possible due to stride value of 6). We observed for linear evaluation setting on the virus dataset the performance dropped from 69.1% to 64.9% (macro-averaged recall), when only same-size targets are predicted. A straight-forward explanation to this is that maximizing the mutual information between the learned embeddings and varying-length target subsequences (both short and long) helps the machine learning model to learn more comprehensive range of semantics and therefore boosts the performance in the downstream supervised tasks.

**Data efficient** We study and compare the performance of Self-GenomeNet with other methods in the low-data regime. We created label-scarcity artificially on the Virus and DeepSEA datasets by subsetting the number of labeled training samples down to 0.1%. We train the models shown in Table 5 using 0.1% of the dataset after pretraining on the whole dataset in self-supervised way. The reported results are when the base networks are fine-tuned. Based on the results reported in Table 5, our method outperforms other methods by substantial margin in the low-data regime.

| Dataset | DeepSEA | | Virus | |
|---|---|---|---|---|
| Metric | *ROC AUC* | *PR AUC* | $Recall_M$ | $F_1$ |
| Supervised | 0.715 | 0.074 | 56.4 | 0.505 |
| CPC | 0.732 | 0.089 | 59.2 | 0.621 |
| Language Model (single nt) | 0.729 | 0.081 | 61.8 | 0.645 |
| Language Model (3-gram) | - | - | 62.9 | 0.634 |
| Contrastive-sc | 0.726 | 0.073 | 61.8 | 0.623 |
| Self-GenomeNet | **0.753** | **0.115** | **67.2** | **0.700** |

Table 5: Comparison of the methods in performance when they are pretrained using the whole dataset in self-supervised manner, followed by supervised training on 0.1% of the dataset. Supervised baseline is not pretrained.

## 7 CONCLUSION

In this paper, we presented Self-GenomeNet, a framework for contrastive learning of nucleotide-level genomic data. Self-GenomeNet is composed of an encoding network followed by a context network and uses a prediction layer for training. The network extracts compact latent representations by predicting representations of the reverse complement of future observations. Self-GenomeNet is optimized end-to-end using unsupervised contrastive loss. We tested the learned representations by linear evaluation, semi-supervised learning, and transfer to other tasks and achieved better performance on the four public bioinformatics benchmarks compared to other self-supervision methods. Moreover, we showed the ability of our method for training in the low-data regime, tested the effect of predicting varying-length target subsequences, and the effect of estimating the output of the context network which reads the predicted subsequence in forward or reverse direction compared to reading reverse-complement of the subsequences.

**Reproducibility Statement:** The source code is available in the supplementary material.

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

## APPENDIX A    PREDICTION NETWORK

**Single prediction network or multiple prediction networks:**    For the pretraining in *DeepSEA* dataset, we trained the network with only one prediction network $q_1$ ($q_m$ where $m = 1$) as well as with multiple prediction networks $m \in \{1, 4, 7, 10, 13, 16\}$ for *DeepSEA*. As mentioned in the manuscript, multiple prediction networks are used when further subsequences are predicted, in addition to the closest subsequence to the predicting subsequence. We report the results of Self-GenomeNet when only one single network is used in the tables 6, 7, 8. Only single prediction network is used during the pretraining stage of the other datasets.

| Dataset | DeepSEA | |
| --- | --- | --- |
| Metric | *ROC AUC* | *PR AUC* |
| CPC | 0.734 | 0.094 |
| Language Model | 0.699 | 0.069 |
| Contrastive-sc | 0.682 | 0.056 |
| Self-GenomeNet - $m \in \{1\}$ | **0.757** | 0.116 |
| Self-GenomeNet - $m \in \{1, 4, 7, 10, 13, 16\}$ | **0.757** | **0.120** |

Table 6: Mean values of ROC AUC and PR AUC metrics for the 919 outputs defined for the *DeepSEA* dataset under linear evaluation protocol.

| Dataset | DeepSEA | | | |
| --- | --- | --- | --- | --- |
| Labeled | 1% | | 10% | |
| Metric | *ROC AUC* | *PR AUC* | *ROC AUC* | *PR AUC* |
| Supervised | 0.776 | 0.119 | **0.865** | **0.239** |
| CPC | 0.780 | 0.125 | 0.863 | 0.236 |
| Language Model | 0.774 | 0.116 | 0.859 | 0.232 |
| Contrastive-sc | 0.764 | 0.105 | 0.848 | 0.222 |
| Self-GenomeNet - $m \in \{1\}$ | **0.795** | 0.135 | 0.862 | 0.234 |
| Self-GenomeNet - $m \in \{1, 4, 7, 10, 13, 16\}$ | **0.795** | **0.138** | **0.865** | 0.235 |

Table 7: Performance of the self-supervision techniques under semi-supervised learning protocol using 1% and 10% of the labels in *DeepSEA* dataset. Supervised baseline without any pretraining is also reported as a baseline. Reported metrics are in line with Table 1.

| Dataset | DeepSEA | |
| --- | --- | --- |
| Metric | *ROC AUC* | *PR AUC* |
| Supervised | 0.715 | 0.074 |
| CPC | 0.732 | 0.089 |
| Language Model | 0.729 | 0.081 |
| Contrastive-sc | 0.726 | 0.073 |
| Self-GenomeNet - $m \in \{1\}$ | 0.750 | 0.103 |
| Self-GenomeNet - $m \in \{1, 4, 7, 10, 13, 16\}$ | **0.753** | **0.115** |

Table 8: Comparison of the methods in performance when they are pretrained using the whole dataset in self-supervised manner, followed by supervised training on 0.1% of the dataset. Supervised baseline is not pretrained.

Here we also report the pretraining time in hours in Table 9 for our method and the baselines on *DeepSEA* dataset using a GeForce RTX 2080 Ti. One can immediately see, the pretraining time for the simplest version of Self-GenomeNet with one prediction network ($m = 1$) is shorter than or comparable to the pretraining time of the other self-supervision methods, whereas the performance is superior.

| Dataset | DeepSEA |
|---|---|
| Method | Pretraining Time (hours) |
| Supervised | 0 |
| CPC | 73 |
| Language Model | 30 |
| Contrastive-sc | 14 |
| Self-GenomeNet - $m \in \{1\}$ | 25 |
| Self-GenomeNet - $m \in \{1, 4, 7, 10, 13, 16\}$ | 158 |

Table 9: Comparison of the methods in performance when they are pretrained using the whole dataset in self-supervised manner, followed by supervised training on 0.1% of the dataset. Supervised baseline is not pretrained.

## APPENDIX B    PROCESSING OF DATASETS AND FILES

For virus-bacteria dataset, data samples are created by reading consecutive samples taken from random regions of FASTA files. Compared to division of FASTA files into same-length non-intersecting sequences, this technique increases number of samples that can be created out of FASTA files. In order to provide randomness, either maximum 90% and 10% of the file is processed for virus files and bacteria files respectively, or maximum 64 data samples are created from a file, before next FASTA file is processed.

For phage/non-phage classification on virus dataset, we set the sequence length to 150. The motivation to limit the input length to 150 nt in some of our evaluations is to be close to a potential application, since this is a typical read length that gets produced during *Next Generation Sequencing* which is frequently used. The length of 150 nt, therefore, allows a direct application of the sequencing output to perform model predictions that can be built using pre-trained models as presented in our work. We also evaluate some experiments with a sequence length of 1,000 nt, because this is a meaningful length of the output of genome assemblers, where these 150 nt long reads are assembled together using overlap information or mapping to reference sequences to produce longer fragments.

The short sequence length can be justified to build neural networks directly on the read output of a sequencer (150 nt), rather than on the output of the assembling process since the latter comes with several limitations. it requires (i) computational resources and often (ii) the presence of *reference genomes*, which are often absent when an environmental sample is sequenced; furthermore (iii) assembling Softwares perform poorly on repetitive regions of bacterial genomes that are of particular interest to study since these often belong to the bacterial defense system. Therefore, traditional bioinformatics software is taking raw read input rather than preprocessed contigs.

## APPENDIX C    OPTIMIZATION OF SELF-SUPERVISED METHODS

For all architectures and all our experiments, we used the same encoder and context models for a fair comparison. Moreover, we took most hyperparameters as given in the original method.

As depicted in Fig 5, we adopted the Henaff (2020) implementation of CPC for 1-dimensional data and use an RNN (LSTM) and a single fully connected layer on top of LSTM layer for prediction as in Oord et al. (2018). It is shown by (Oord et al., 2018) that the performance for the baseline models decreases significantly when the span of predicted future observations is too short, however it decreases slightly when they are too long. To this extend, the predicted span for the patches are selected to be not too short as well as not too long compared to the full length of the sequence. For pretraining on the virus dataset, every second feature from $4^{th}$ to $10^{th}$ derived by encoder network is predicted with different linear layers. Thus, the representations derived from the following 60 nucleotides are predicted, when the full sequence length is 150. For bacteria pretraining, the following $2^{nd}$ to $10^{th}$ features are predicted, making the prediction span of 200 nucleotides for the sequence length of 1000 nucleotides. For the DeepSEA dataset, every second feature from $3^{rd}$ to $20^{th}$ features are predicted, making the span up to 260 nucleotides for the sequence length of 1000.

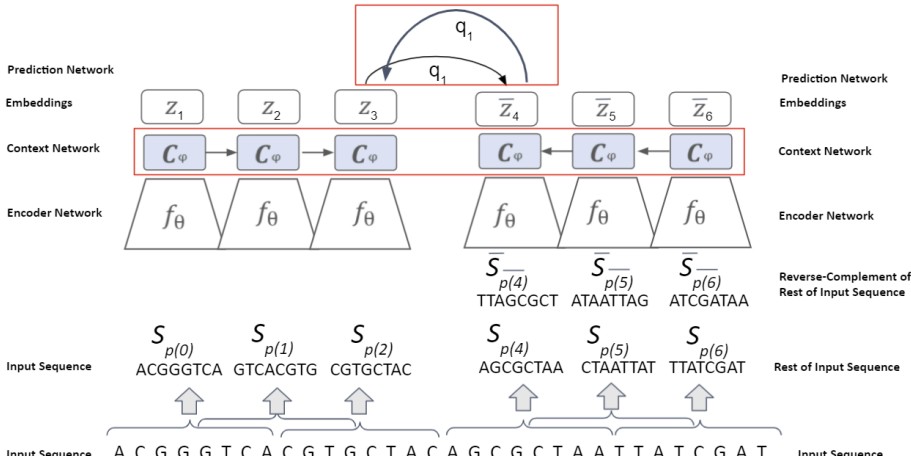

Figure 3: *How is Self-GenomeNet inherently symmetrical?* The context network and the prediction network with the shared weights are used both for predicting the subsequence on the right from the left one as well as the left subsequence from the the right one.

For Contrastive-sc (Ciortan and Defrance, 2021) (Fig. 5), we used the masking rate of 0.9 as recommended by the paper and did not use additional linear layer on top of the final outputs of LSTM.

For the language modelling (Fig. 4), we followed different strategy for different datasets. For the virus and bacteria dataset, the model predicts the following 60 and 200 nucleotides respectively, which are taken from outside of the sequence. For DeepSEA, the nucleotides outside the sequence is not provided within the dataset. Thus, the last 260 nucleotides are estimated from the first 740 nucleotides during pretraining. Please note that the span of predicted observations is the same for the CPC and the language modelling.

In the supervised training of virus datasets, we firstly train only the final linear layer while the pretrained base model is frozen. Once this layer is optimized, starting from this point, we tune the whole network. For DeepSEA dataset, we directly optimize the whole model for both freezed and unfreezed base-networks.

## C.1 ADDITIONAL EXPERIMENTAL RESULTS

In this section we further compare our proposed Self-GenomeNet with the baseline methods for linear evaluation (Fig. 6), semi-supervised learning (Fig. 8, Fig. 9), transfer learning (Fig. 10,Fig. 11), and data efficiency (Fig. 7).

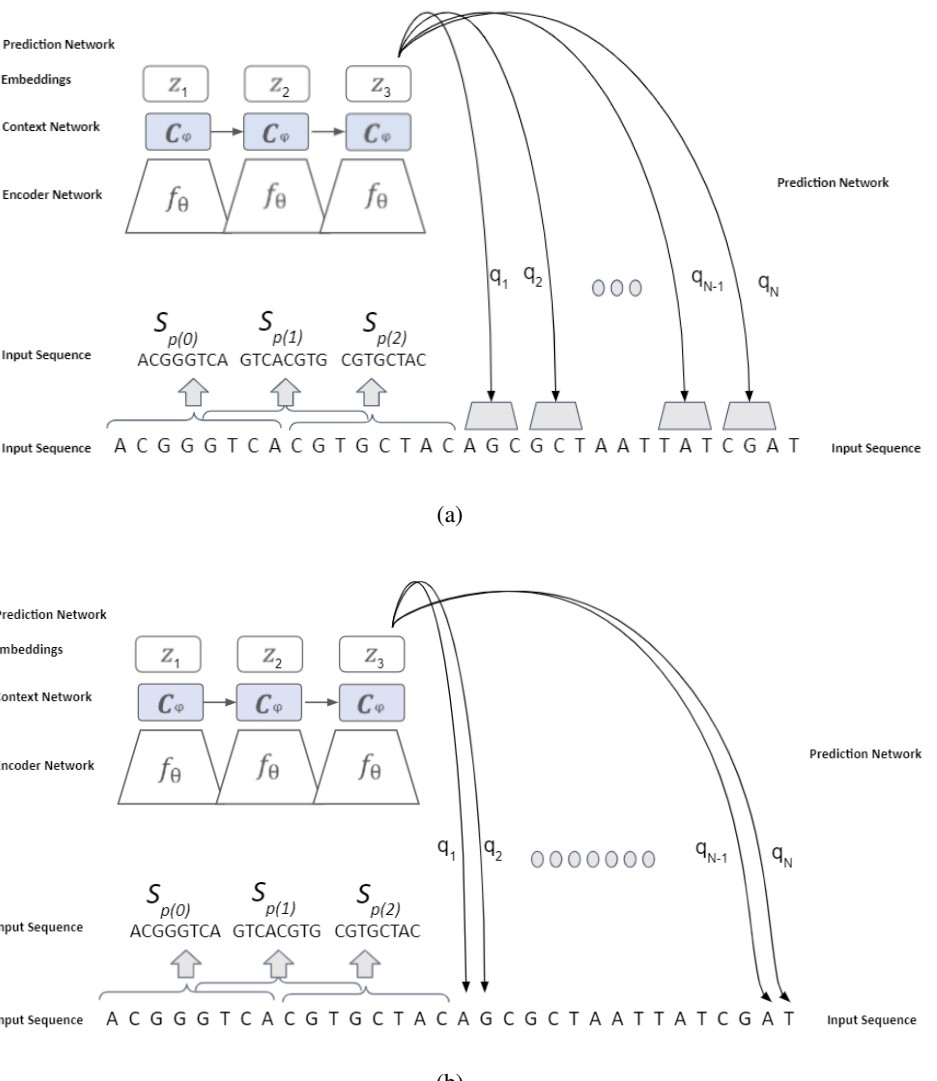

(a)

(b)

Figure 4: Illustration of Language Model using 3-gram as targets (on the top), followed by Language Model using single nucleotides as targets (in the bottom); In language modeling, future nucleotides are predicted using cross-entropy loss. In (a) the nucleotides are one-hot encoded as a group of 3 nucleotides into a vector of 64, whereas in (b) each nucleotide is one-hot encoded seperately as a vector of 4.

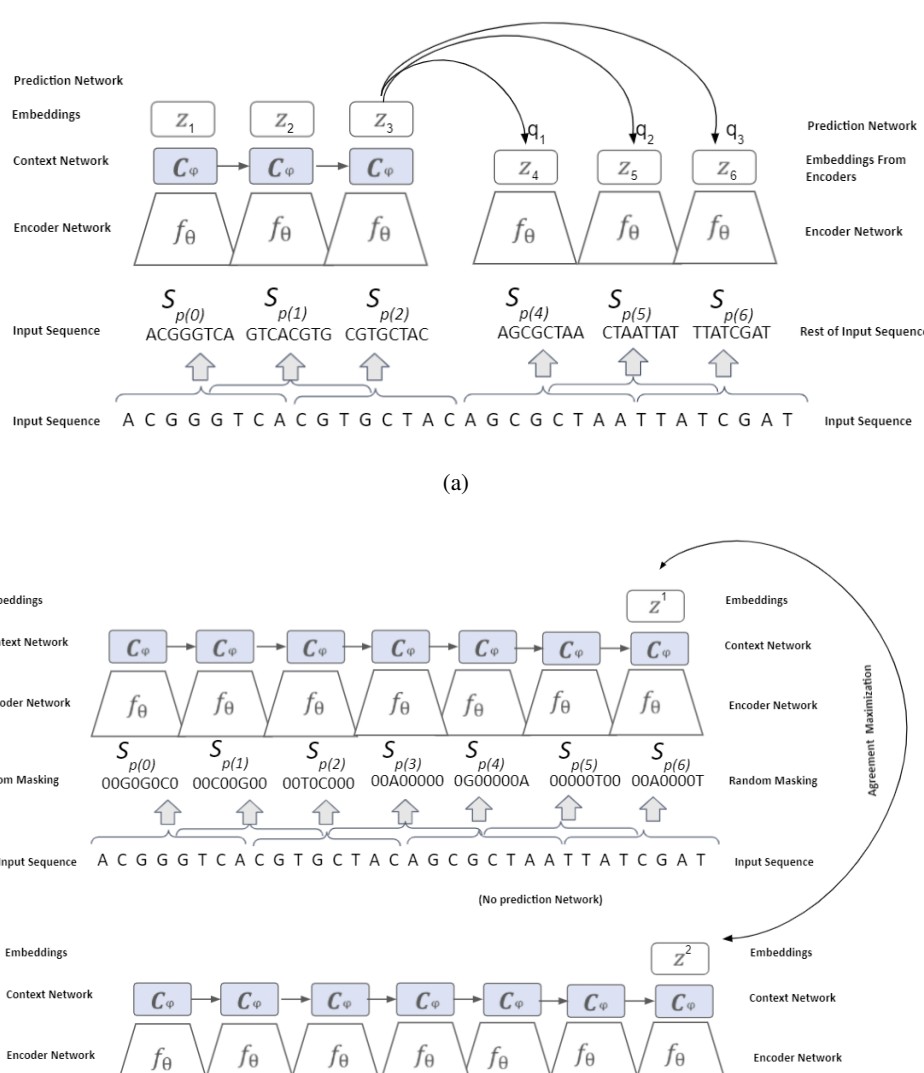

(a)

(b)

Figure 5: Illustration of CPC (on the top) and Contrastive-sc (in the bottom); CPC predicts the output representations obtained from the encoder network for the future patches using the contrastive loss, where the negative samples are representations of other patches. Contrastive-sc takes a sequence and copy of sequence and mask the input sequence randomly. Then, the agreement between embedding of two masked copies are maximized using the contrastive loss, where negative samples are embeddings of other masked sequences.

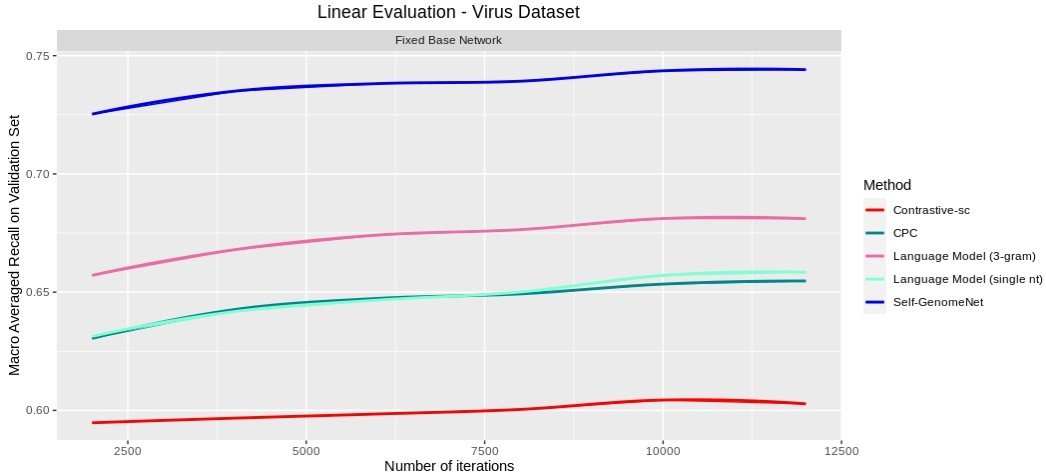

Figure 6: Linear evaluation

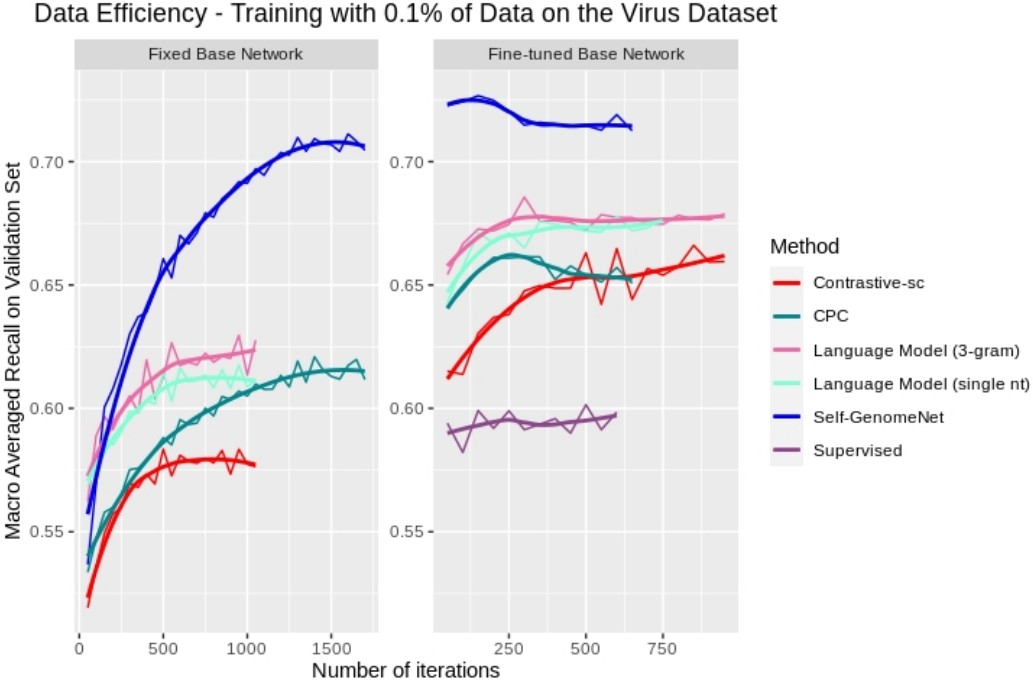

Figure 7

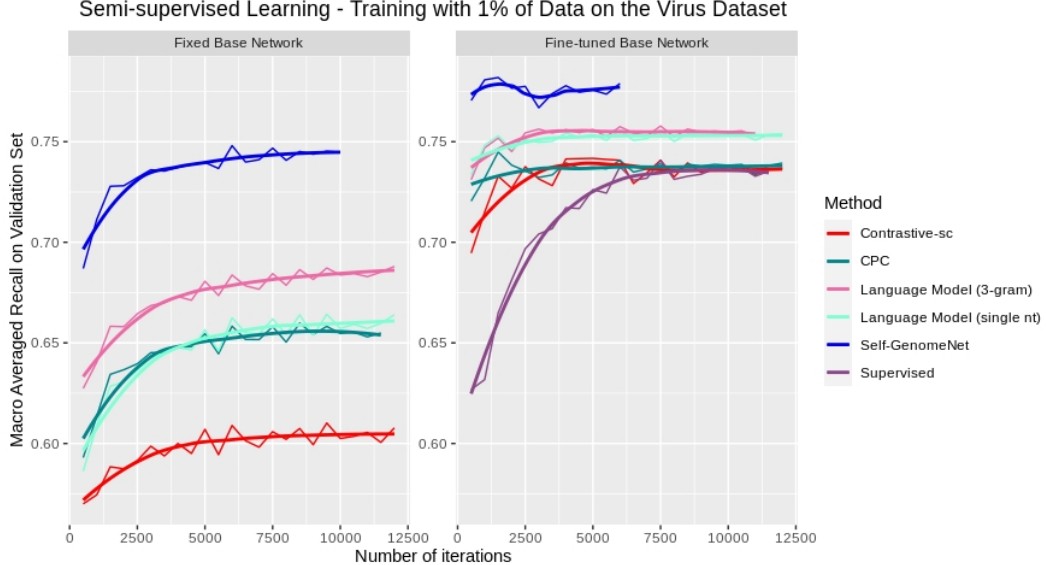

Figure 8: Semi-supervised learning

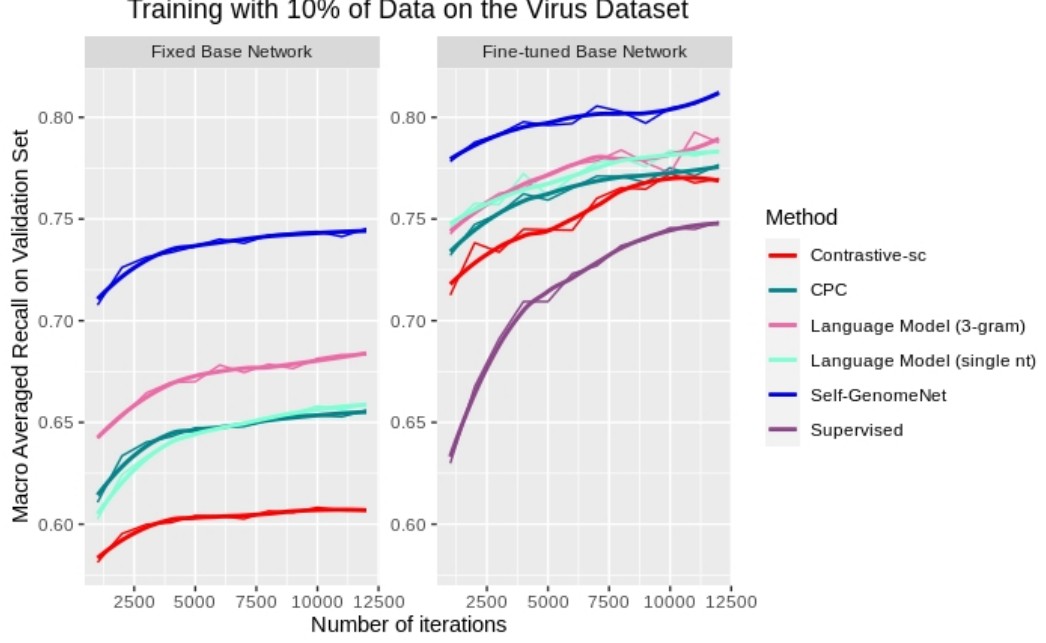

Figure 9

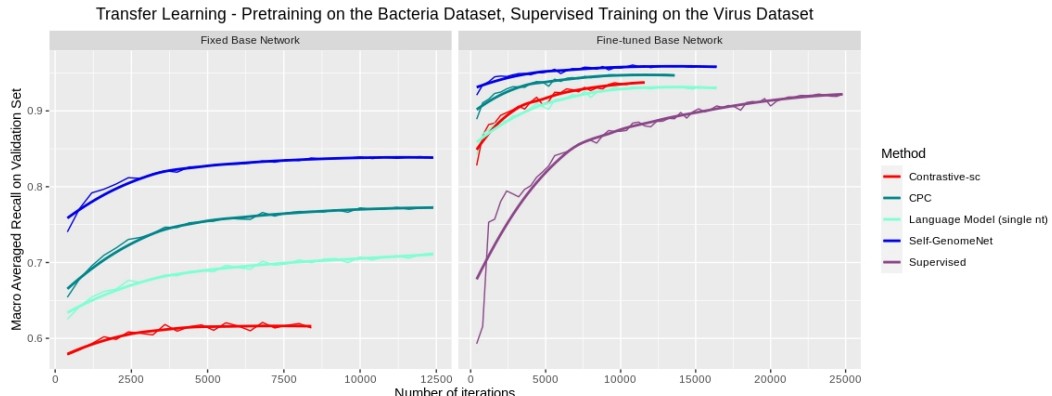

Figure 10

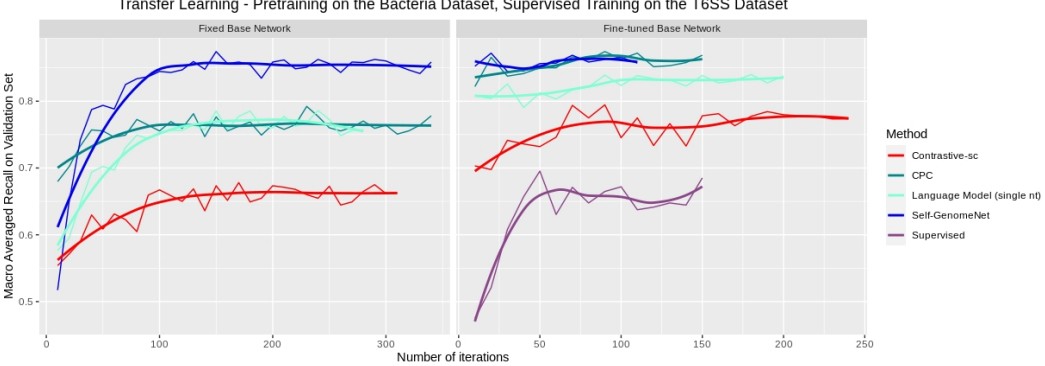

Figure 11: Transfer Learning

