# OpenReview forum: "Self-GenomeNet: Self-supervised Learning with Reverse-Complement Context Prediction for Nucleotide-level Genomics Data"
_ICLR.cc/2022/Conference — ICLR 2022 Submitted_

### Official Review · Reviewer_9TG8 · 2021-10-23

**Correctness:** 4
**Technical Novelty And Significance:** 2
**Empirical Novelty And Significance:** 3
**Recommendation:** 5
**Confidence:** 4

**Main Review:**

**Strengths**
* The proposed method is simple yet effective. The domain-specific data augmentation (e.g., reverse complement) improves the representation power considerably.
* Unlike CPC or skip-gram methods that use a fixed-length short subsequence, the proposed method trains models based on varying length sequences.
* The improvement compared to baselines CPC, Language model, and even supervised learning method is considerable.


**Weaknesses**
* The proposed method is constructed by existing methods and beyond the genome data, the benefits of the proposed method are limited.
* If the authors provide more explanation about the difference between the proposed method and baseline algorithms such as contrastive-sc, it would be better to evaluate the main contributions of this manuscript.

**Summary Of The Paper:**

The authors proposed a self-supervised learning method for nucleotide-level genomic data utilizing reverse-complement of genomic sequences. The proposed method achieved a considerable performance improvement. In addition, the authors proposed an architecture called Self-GenomeNet that handles varying-length genome sequences.

**Summary Of The Review:**

The proposed method is a simple and effective contrastive learning method and shows great representation power. The proposed method showed the best performance in many different settings such as linear evaluation and semi-supervised learning. The reverse-complement based data augmentation seems simple and effective. However, the proposed framework is overall a simple combination of existing methods and beyond genome datasets, the impact of this proposed method is questionable.

---

> ### Author Response · Authors · 2021-11-19
> **Response to Reviewer 9TG8**
>
> We thank the reviewer for the useful comments and suggested improvements.
>
>
> ### The impact of Self-GenomeNet:
> > * The proposed method is constructed by existing methods and beyond the genome data, the benefits of the proposed method are limited.
>
>  While the translational value of our work to other DL domains such as image recognition is limited, the value that optimized learning regimes and models could bring to human prosperity are hard to overstate and much more obvious than potential applications of more studied data modalities and DL applications such as text to image synthesis. Just to name a few: neural networks optimized for genomic data modalities could benefit outbreak detection due to the optimization of taxonomic assignments (Nissen et al. 2021) and could contribute to the design of novel therapeutica due to the modeling of protein folding (Jumper et al. 2021).
>
> ### References
>
> * Jumper, John, Richard Evans, Alexander Pritzel, Tim Green, Michael Figurnov, Olaf Ronneberger, Kathryn Tunyasuvunakool, et al. 2021. “Highly Accurate Protein Structure Prediction with AlphaFold.” Nature, July. https://doi.org/10.1038/s41586-021-03819-2.
>
> * Nissen, Jakob Nybo, Joachim Johansen, Rosa Lundbye Allesøe, Casper Kaae Sønderby, Jose Juan Almagro Armenteros, Christopher Heje Grønbech, Lars Juhl Jensen, et al. 2021. “Improved Metagenome Binning and Assembly Using Deep Variational Autoencoders.” Nature Biotechnology 39 (5): 555–60.
>
>
>
> ### Comparison with the baseline methods:
> > * If the authors provide more explanation about the difference between the proposed method and baseline algorithms such as contrastive-sc, it would be better to evaluate the main contributions of this manuscript.
>
> We improved the writing of the manuscript and added additional descriptions about the baseline algorithms in Section 2, *Self-supervised learning* and *Self-supervised learning in bioinformatics*.
> In Section 5, we clarified our motivation for comparison with the baselines. We also added some new Figures in the appendix to compare the learning procedure of our method with the compared self-supervised baselines.
>
> (edited for formatting)

---

> > ### Author Response · Authors · 2021-11-29
> > **Further Comments for Reviewer 9TG8**
> >
> > We would like to use this opportunity and clarify some points further as today is the last day of the rebuttal period. In particular, regarding the main contributions of this manuscript:
> >
> > ### Comparison with the baseline methods and Contributions:
> >
> > > * If the authors provide more explanation about the difference between the proposed method and baseline algorithms such as contrastive-sc, it would be better to evaluate the main contributions of this manuscript.
> >
> > >  [...] However, the proposed framework is overall a simple combination of existing methods [...]
> >
> > Regarding the differences between our method and other previous methods, we provide some figures in Appendix (page 15-16 of the manuscript), where we visualized the self-supervised baseline algorithms (Fig. 4 (a,b) and Fig. 5 (a,b)).
> >
> >
> > **Differences to CPC**
> >
> > Very briefly, the CPC predicts embeddings of the target subsequences, calculated by the encoder network in a contrastive manner. While our method similarly predicts representations of subsequences contrastively, it predicts the representation of the output of the context network applied to this subsequence. This means that the predicted representations are of sequences that are (i) considerably longer than in CPC, spanning multiple sequence patches, and (ii) have varying length, unlike fixed-length patches in CPC. Here we make use of the reverse-complement symmetry in genome sequences to efficiently build multiple varying length representations at once. Other methods make use of the RC in different ways (for example using both forward and RC as input at the same time, or using the RC as a data augmentation method), but to our knowledge, the way in which we exploit RC in our method is novel as well (besides points (i) and (ii) listed above).
> >
> > **Differences to Contrastive-sc**
> >
> > The Contrastive-sc predicts the output of the context network for a partially and randomly masked copy of the sequence in a contrastive manner from the output of the context network for another partially and randomly masked copy of the sequence. While this method *does* compute a contrastive loss of two sequences of the same length (as opposed to CPC), it is still considerably different from our method. First, our method only considers non-overlapping subsequences, Contrastive-sc uses random masking of input data instead. Second, our method compares subsequences of varying lengths, whereas all subsequences compared in contrastive-sc have the same length.
> >
> > **Novelty of RC symmetry**
> >
> > We now stress the point that we are using the *RC symmetry* in novel ways, different than other methods, in the first paragraph of Section 3.2 (very end of page 4 to the beginning of page 5):
> > > The entire procedure is symmetric since our model learns sequence and the RC of sequence. [...] In other words, with the help of the symmetry induced by RC, substantial weight sharing is provided.

---

> > > ### Comment · Reviewer_9TG8 · 2021-12-03
> > > **Thanks for the author feedback.**
> > >
> > > I thank the authors for the detailed feedback. Now the contribution of this paper becomes clearer.

---

### Official Review · Reviewer_ERw2 · 2021-10-27

**Correctness:** 2
**Technical Novelty And Significance:** 2
**Empirical Novelty And Significance:** 2
**Recommendation:** 5
**Confidence:** 4

**Main Review:**

-	While the paper has its own merits, my biggest concern is that the connection between motivation and the proposed method is not crystal clear. I agree that pursuing the representation equivariance of two reverse-complementary sequences is an important research direction. However, the proposed method deals with representations of two subsequences. Even if they are originally from a single sequence, RC of one subsequence does not make it reverse-complementary to the other subsequence. I’m not sure why the prediction of S^bar_(N:t’) is more helpful than the prediction of S_(N:t’) or S^bar_(1:N).
-	On page2-line3, the authors stated that “our method divides a sequence into two complementary subsequences..” I don’t think the authors meant that they are composed of complementary nucleotides (A^bar = T), but it might be misleading for some readers.
-	On page3-line10, the authors stated that “positive pairs include the input sequence and a reverse-complement of the sequence.” However, this is not entirely correct. The proposed method used one subsequence and the RC of the other subsequence as the positive pairs.
-	On the page3-last sentence of the background, the authors stated that “exploit this genome-specific feature to tighten the lower bound between mutual information between learned representations.” However, I could not find theoretical or empirical grounds to claim that the proposed method tightens the mutual information lower bound. Please clarify the point.
-	In Sec3.2-line4, the authors stated that the proposed architecture makes training more efficient. Can you explain why it is more efficient?
-	Please provide some missing experiment setups. (1) which value was used for “m” in the proposed loss term. (2) length statistics of the three datasets. (3) Training time and learning curve for the proposed method.
-	According to the network architecture section, it seems using the DanQ model architecture for the DeepSEA dataset causes some issues that only 3.6% of parameters are trained in the self-supervised stage. Then, why didn’t you use the model architecture used for the Virus dataset?
-	To be self-contained, please provide more algorithmic details for the compared baselines. In addition, can you clarify whether the baselines use the N-gram or single nucleotide for the pre-training? If it is the latter, can you also provide the results for using the N-gram?
-	Do you have any specific reasons for using the recall as the evaluation metric? Can you also compare them in terms of the F1 score?
-	In Table7, the authors compared the autoregressive models reading the target subsequences in forward, reverse, and RC directions. What is the difference between the forward model and the CPC model? In my view, the relatively small performance difference between the forward and RC models might indicate that the effectiveness of the proposed method based on the RC-equivariance is not significant.
-	While the proposed method is largely motivated by the RC equivariance, the paper does not seem to provide sufficient theoretical or empirical analyses concerning the point. (1) Since the RC-equivariance is not guaranteed within the model, can you show how well the learned representations satisfy the RC equivariance? (2) Can you compare its performance with those of previous RC-equivariance techniques on some NGS datasets?


**Summary Of The Paper:**

Recently several works have proposed semi-supervised learning methods to leverage unlabeled biological sequences for learning their general-purpose representations. In this work, the authors proposed the Self-GenomeNet, a novel contrastive learning method for nucleotides based on the reverse-complement (RC) context prediction. First, given a sequence, they divide it into two subsequences and transform one into its RC. Then, the model is trained to distinguish their representations from those of other random nucleotide sequences. The authors claimed that the proposed method considerably outperforms previous self-supervised baseline models on three benchmark datasets in both self-supervised and semi-supervised evaluation

**Summary Of The Review:**

While the paper has its own merits, it has several issues to be addressed regarding (1) the connection between motivation and the proposed method, (2) theoretical or empirical analyses concerning the RC-equivariance, and (3) some questionable experiment setups.

---

> ### Author Response · Authors · 2021-11-19
> **Response to Reviewer ERw2 (1/2)**
>
> We thank the reviewer for the useful comments and suggested improvements. We addressed minor issues and modified the manuscript.
>
> ### Motivation of the proposed method :
> > * While the paper has its own merits, my biggest concern is that the connection between motivation and the proposed method is not crystal clear. [...] I’m not sure why the prediction of S^bar_(N:t’) is more helpful than the prediction of S_(N:t’) or S^bar_(1:N).
>
> Thank you for the question; we first answer why the prediction of S^bar_(N:t’) is more helpful than the prediction of S_(N:t’):
>
> While it is common to use the same model for both a sequence and its reverse-complement in RC-equivariant architectures, we use the RC to induce symmetry for self-supervised pre-training for the first time. When the model predicts the representation of S_(N:t') without complement from S_(1:t) (the "*reverse*" condition in our ablation study) the context network reading S_(N:t') backward and the prediction network of that subsequence should ideally not use the same weights of the context network or prediction network reading S_(1:t) in the forward direction. However, it is legitimate to use the same context and prediction networks when RC is used. Since both RC and forward subsequences are “valid”, the same machine learning model can be used for both of them.
>
> > * and here is our answer for the second part [why the prediction of S^bar_(N:t’) is more helpful than S^bar_(1:N).]
>
> Predicting S^bar_(1:N) is a simple task for a machine learning model to predict from S_(1:t), given that there is a direct relationship between the S_(1:t) and S^bar_((N-t+1):N). Therefore, given sufficient data, the model would just memorize this relationship and would not infer the global semantics of the sequence, which would be helpful for learning better representations for the supervised downstream task.
>
> We might have not made this reasoning clear enough in the manuscript, and have therefore extended the first paragraph of Section 3.2.
>
>
> > * In Sec3.2-line4, the authors stated that the proposed architecture makes training more efficient. Can you explain why it is more efficient?
>
> In our paper, we show that the prediction of the output of the context network increases the accuracy when varying-length targets are used. When we feed both S_(1:N) and S^bar_(N:1) into the network, then the context network (LSTM) creates the representations for S_(1:t) and S^bar_(N:t') for many t and t' efficiently (since the LSTM needs to compute the output of t to get the output for t+1 in any case). This makes it possible to compute loss for many combinations of different (t,t')-pairs in a single batch computation step. Using S_(N:1) instead of S^bar_(N:1) would still make this optimization possible, but it would be necessary to either have twice as many network weights or to represent invalid sequences (see above). However, much worse, if the forward sequence were used for both subsequences that are compared, then it would be necessary to calculate the outputs for the LSTM applied to S_(t':N) for different values of t' from scratch, either by using independent LSTM layers for each value of t', or by cycling through different single t'-values for each batch. (The same would be the case if the context network were not an LSTM but anything else, except a recurrent unit with reversed direction, which would again have the same problem as the first case). Using S_(1:t) and S^bar_(t':1) as subsequences to compare against has the unique advantage of making this optimization possible.
>
> We have also added a small reference to these facts to the first paragraph of Section 3.2, and we can expand it more if you think this is an important point.
>
> > * While the proposed method is largely motivated by the RC equivariance, the paper does not seem to provide sufficient theoretical or empirical analyses concerning the point. (1) Since the RC-equivariance is not guaranteed within the model, can you show how well the learned representations satisfy the RC equivariance?  (2) Can you compare its performance with those of previous RC-equivariance techniques on some NGS datasets?
>
> We neither claim our method provides RC-equivariance nor do we think our method is really comparable to previous RC-equivariance techniques. As we clarified in the previous questions our method makes use of RC in order to provide several advantages. Nevertheless, RC-equivariance techniques and our method are not mutually exclusive and these techniques might potentially further increase the performance. We changed the wording on some parts that may have been misleading before to stress that in our method, we use RC in order to create symmetry within our self-supervision technique, which provides several advantages such as parameter sharing for the context and prediction networks.

---

> > ### Author Response · Authors · 2021-11-19
> > **Response to Reviewer ERw2 (2/2)**
> >
> > ### Evaluation Metric
> > > *  Do you have any specific reasons for using the recall as the evaluation metric? Can you also compare them in terms of the F1 score?
> >
> > For the virus dataset, we had chosen macro average recall as it is easy to interpret since this metric is equivalent to class-balanced accuracy. However, we now additionally report the F1 metric for experiments on the virus dataset besides macro-averaged recall.
> >
> >
> > ### N-Gram
> > > * To be self-contained, please provide more algorithmic details for the compared baselines. In addition, can you clarify whether the baselines use the N-gram or single nucleotide for the pre-training? If it is the latter, can you also provide the results for using the N-gram?
> >
> >
> > As for the sequences, the predicting subsequence consists of single nucleotides. The predicted part of the nucleotides are also single nucleotides, however just like a representation is formed by a machine learning model for predicting subsequence, very briefly, CPC predicts embeddings calculated by the encoder network and contrastive-sc predicts the output of context network for a partially masked copy of the whole sequence in a contrastive manner. On the other hand, language models predict single nucleotides in our previous experiments whereas it could potentially predict N-grams. We will provide results of language modeling predicting N-gram representations on the virus dataset until the end of the rebuttal phase. We implemented the method and it’s under pre-training now. We will not be able to provide results for N-gram for other experiments until the end of the rebuttal phase, 22/11/21, due to the expensive computational resources (such as the long training time) that they need. Additionally, we provided some figures where we compare the baseline models in the appendix for better visualization.
> >
> >
> > ### Significance of the Ablation Study Results
> > > * In Table7, the authors compared the autoregressive models reading the target subsequences in forward, reverse, and RC directions. What is the difference between the forward model and the CPC model? In my view, the relatively small performance difference between the forward and RC models might indicate that the effectiveness of the proposed method based on the RC-equivariance is not significant.
> >
> > We have recently performed new experiments for Forward, Reverse, and Reverse-complement strands. For each, we pretrained the models in self-supervised fashion three times and then trained the pretrained models in supervised way under linear evaluation protocol.
> >
> > | Forward (F1)  |  Reverse(F1)  |  RC (F1) |
> > |  --------- | ---------| ----------- |
> > |  0.7048$\pm$0.0025  | 0.6952$\pm$0.0003  |  0.7089$\pm$ 0.0008 |
> >
> >
> >
> >
> > ### Other Minor Issues
> > We thank you for pointing us towards some of the minor issues. We have used this opportunity to clean up the manuscript, fix the mentioned and other typos. In particular, we now use the word *complementary* to only refer to complementary nucleotides (Section 1), and we are now more specific about the fact that positive pairs are a subsequence and the RC of the remaining sequence. We elaborate more about **the theoretical** connection between the employed contrastive loss and the mutual information by adding two paragraphs about this in Section 3.2. We have also provided missing values ("m", dataset statistics). And will provide the curves before the end of the rebuttal phase.

---

> > > ### Author Response · Authors · 2021-11-29
> > > **Further Comments for Reviewer ERw2**
> > >
> > > We want to use this opportunity to respond to a few comments that we may not have addressed adequately before:
> > >
> > > ### Connection between Motivation and Method
> > > > * While the paper has its own merits, my biggest concern is that the connection between motivation and the proposed method is not crystal clear [...].
> > >
> > > We hope we have also addressed this point with the addition we made to our manuscript (submitted on Nov. 22nd) in the first paragraph of section 3.2; it starts at the last quarter of page 4:
> > > > We aim to produce an embedding of a subsequence $S\_{1:t}$ that contrastively predicts the embedding of the RC of the remaining subsequence, $\overline{S}\_{N:t'}$ for some $t' > t$. [...] In other words, with the help of the symmetry induced by RC, substantial weight sharing is provided.
> > >
> > > ### Choice of Model Architecture and Baseline
> > > > * According to the network architecture section, it seems using the DanQ model architecture for the DeepSEA dataset causes some issues that only 3.6% of parameters are trained in the self-supervised stage. Then, why didn’t you use the model architecture used for the Virus dataset?
> > >
> > > DanQ is a standard model for the DeepSEA dataset. Many other papers use it as a baseline for this dataset (Tayara et al. 2019, Zhang et al. 2019). By using DanQ, we demonstrate that our method works well as a self-supervised method when used with previously developed models for the given dataset and task. Even though optimizing the used machine learning model might further increase the achieved performance, we believe optimizing the model architecture is beyond the scope of this paper as the main contribution of the paper is the proposed self-supervision technique. In our paper, we show that our method works considerably better than the baseline self-supervised methods for several datasets and tasks.
> > >
> > > **References:**
> > > * Tayara, Hilal, and Kil To Chong. "Improving the quantification of DNA sequences using evolutionary information based on deep learning." Cells 8.12 (2019): 1635.
> > > * Zhang, Hanyu, et al. "NCNet: Deep learning network models for predicting function of non-coding DNA." Frontiers in genetics 10 (2019): 432.
> > >
> > > ### Difference between "Forward" model and CPC model
> > >
> > > > * In Table7, the authors compared the autoregressive models reading the target subsequences in forward, reverse, and RC directions. What is the difference between the forward model and the CPC model?
> > >
> > > The CPC model predicts the output of the *encoder network*, whereas what we term the "forward" model predicts the output of the *context network*. We now have the following text in our paper:
> > > > In *forward* setting, the context network reads the target subsequence $S\_{t':N}$ in the same direction as it reads the predicting subsequence $S\_{1:t}$
> > >
> > > The target to be predicted contrastively in our "forward"-model is therefore the representation of the subsequence $S\_{t':N}$; in CPC, only the representations of individual patches (as encoded by the encoder network) are predicted. The effective consequence of this is that our "forward" model predicts (i) longer subsequences, and (ii) varying-length subsequences (since $t'$ varies between batches). The "forward" model is therefore a part of our ablation study where we investigate the effect of each component by only changing that component (whether to predict RC or forward sequence) and keeping the rest of our method as is (i.e. the differences from CPC as listed above).
> > >
> > > ### Training plots, training time, and “m” value in the loss function
> > > > * which value was used for “m” in the proposed loss term. [...] Training time and learning curve for the proposed method
> > >
> > > The detailed information regarding the “m” value in the loss function is now explained in Appendix A and listed in Tables 6-9 (pages 12-13). See Table 9 (page 13) for training times. We also plot learning curves in Appendix C.1, figures 6 to 11 (pages 17-19).

---

### Official Review · Reviewer_4DkC · 2021-11-02

**Correctness:** 3
**Technical Novelty And Significance:** 4
**Empirical Novelty And Significance:** 3
**Recommendation:** 8
**Confidence:** 4

**Main Review:**

The question of self-supervision for networks predicting from DNA is
timely, given the success encountered by this paradigm in other
applications of learning. The proposed contribution is novel, and does
seem to improve the prediction performance on a variety of tasks. The
evaluation is thorough, including large datasets from different
domains of the tree of life, relevant baselines and assessments on
fine-tuning, semi-supervised and transfer learning settings. It also
includes an ablation study confirming in particular the contribution
of (i) training on variable sequence sizes and (ii)
reverse-complementing the piece of the sequence to be predicted.

My main concern is on the clarity of the manuscript, in particular:

- The terms "encoder" and "context" networks are used without being
  clearly defined. It could help to say earlier to which architectures
  (respectively, a CNN and an RNN) they refer specifically.

- I found the use of macro-averaged recall confusing: my understanding
  is that in this context (binary classification) it is equivalent to
  balanced accuracy, but it first seems like only recall is considered
  (as opposed to eg precision or anything else quantifying false
  positives).

- The claims on RC-invariance should be clarified. For example in 3.2
it is claimed that "The architecture enforces the symmetry implied by
the RC". Is it really the case? It seems like the network could learn
to be RC-invariant, but that the invariance is not built in.

- It was not immediately clear to me why the self-supervision task
made sense (predicting the end of a sequence from its start,
contrastively to other sequences). It made more sense when I saw the
task of predicting a class of sequence (prophage vs non-prophage
viruses. Maybe this could be justified earlier. Maybe more
importantly, is the relevance of this self-sueprvision universal, or could it be
less efficient on some other tasks?

The writting could also be improved, eg:

- "One likely explanation is that, because we maximized the mutual
information between the learned embeddings and varying-length target
sequences with both short and long inputs."

- "One loss term is introduced for each index i, denoting
  the number of patches represented by z i"

- "We investigated the effect of the change in length of target subsequen-
ces by comparing our method against the single-length or longer target subsequences."

- "Self-GenomeNet uses the embedding representation of a context
network for downstream tasks which learns from reverse complement as a
target for its predictions."

- In Section 3 (motivation) I also found the formulation unclear: the
text suggests that CPC doesn't deal with multiple sizes, and then says
the opposite.


**Summary Of The Paper:**

This submission introduces self-genomenet, a self-supervised training
method for learning from DNA sequences. The self-supervision is done
by predicting the end of a sequence from its start (both broken into
smaller subsequences), through a contrastive loss against other random
sequences. The predicted part is also reverse-complemented (RC),
making the network learn the expected reverse-complement invariance of
the prediction function. The method is extensively tested on several
learning tasks, where it shows good performances.

**Summary Of The Review:**

The contribution is novel, timely and useful to the community. Its assessment is thorough. The clarity of the manuscript could be improved.

---

> ### Author Response · Authors · 2021-11-19
> **Response to Reviewer 4DkC**
>
> We thank the reviewer for the encouraging words, useful comments, suggested improvements, and good score. We improved the manuscript and explained more the motivation of our proposed method in Section 3.1 and Section 3.2.
>
>  ### Evaluation Metric:
> > * I found the use of macro-averaged recall confusing: my understanding is that in this context (binary classification) it is equivalent to balanced accuracy, but it first seems like the only recall is considered (as opposed to eg precision or anything else quantifying false positives).
>
> Thank you for pointing this out. For binary classification, if we would assume one class is considered as a positive sample, macro-average recall is evaluated as (TP/(TP+FN)+TN/(TN+FP))/2 which is equivalent to the class-balanced accuracy. Therefore, not only recall is considered in the classical sense where true negative or false positive samples do not have an impact on the score. Nevertheless, for better consistency, we also calculated the F_1 score for all the experiments and updated our manuscript.
>
> ### Motivation of proposed method :
> > * The claims on RC-invariance should be clarified. For example, in 3.2 it is claimed that "The architecture enforces the symmetry implied by the RC". Is it really the case? It seems like the network could learn to be RC-invariant, but that the invariance is not built-in.
>
> We do not claim our resulting representation is RC-invariant; however, the self-supervised training process is. This is because both the first subsequence $S\_{1:t}$ is used to contrastively predict $S\_{N:t'}$ against alternatives, but $S\_{N:t'}$ *also* predicts $S\_{1:t}$ against alternatives. Although the method might learn RC-aware representations to some extend, we mainly use reverse-complement property to produce symmetry in design and therefore to be able to use shared weights for the context and prediction networks in a reasonable manner. We modified the following passage to the text:
> We aim to produce an embedding of a subsequence $S_{1:t}$ that contrastively predicts the embedding of the RC of the remaining subsequence, $ \overline{S}\_{N:t'}$ for some t' > t. Predicting $\overline{S}\_{N:t'}$, i.e. the RC of $S\_{t':N}$, instead of just the reverse of $S\_{t':N}$ , has the benefit that it is possible to use the same embedding and context network for both the first and the second subsequence, as $\overline{S}\_{N:t'}$ is a valid genome sequence, whereas $S\_{N:t'}$ is not. We use a recurrent layer to build the representations of both S and $\overline{S}$, so representations of different length subsequences are calculated at once by using the hidden state of the recurrent unit at different points in the sequence. This makes it possible to contrastively predict subsequences of varying lengths against each other efficiently in a single training step. This has an advantage over using two forward sequences $S\_{1:t}$ and $S\_{t':N}$, where using different starting point values t' in one step would be much more computationally demanding since the values of recurrent units would have to be calculated fresh for every t'. The entire procedure is symmetric by design thanks to the RC property of the sequences. This symmetry provides the possibility to use a context network as well as the prediction networks with shared weights for both when reading the subsequence in forward direction as in the left subsequence in Fig. 2 as well as when reading the reverse-complement of the sequence on the right.
>
> > * It was not immediately clear to me why the self-supervision task made sense (predicting the end of a sequence from its start, contrastively to other sequences). It made more sense when I saw the task of predicting a class of sequence (prophage vs non-prophage viruses. Maybe this could be justified earlier. Maybe, more importantly, is the relevance of this self-supervision universal, or could it be less efficient on some other tasks?
>
> Self-GenomeNet showed outstanding performance on different experimental settings, learning tasks, and datasets. We performed three different downstream tasks (we now have also included an experiment regarding the identification of T6SS effector proteins). More importantly, we believe the virus and t6SS effector protein identification differ from the DeepSEA task in an important way, as it is trying to differentiate an aspect of an “entire given genome” sequence such as a taxonomical classification or protein type, whereas for DeepSEA dataset “regions within a sequence” might be playing a role to decide on the label of the dataset rather than the whole sequence. We, therefore, estimate the relevance of our method is universal.
>
> ### Other minor issues
> We thank you for pointing us towards some of the minor issues. We have used this opportunity to clean up the manuscript, fix the mentioned and other typos.

---

> > ### Comment · Reviewer_4DkC · 2021-11-26
> > **Acknowledgement of the rebuttal**
> >
> > I thank the authors for their answer, it adequately addresses my concerns.

---

### Official Review · Reviewer_ifj3 · 2021-11-03

**Correctness:** 3
**Technical Novelty And Significance:** 2
**Empirical Novelty And Significance:** 3
**Recommendation:** 6
**Confidence:** 4

**Main Review:**

Strengths:

+ The idea to use self-supervised contrastive learning for genomic sequences is interesting and well-motivated
+ Proposed changes to the existing model are domain-specific and relevant for the application of the method for genomic tasks
+ The improvement in classification performance over supervised models is promising, especially for 0.1% and 1% labels cases, is promising.
+ The results indicate that learning robust representations via contrastive loss-based self-learning of the sequences can lead to better performances for supervised and semi-supervised learning in genomics.

Weaknesses:

- The paper's central claim is that self-learning-based methods can help with genomics tasks with limited data. However, the datasets used in the paper do not reflect those tasks and have an ample amount of labeled information. While the label scarcity is simulated in some results, it would be helpful to demonstrate the applicability of Self-GenomeNet for classification for genomic tasks with limited labels.
- The rationale for the choice of the datasets, models, and baselines is not entirely clear. For example, there are more recent methods (like Basenji) that perform genome sequence classification. So why was DanQ chosen as the model of choice?
- Similarly, how was the model architecture for classifying viral genomes selected? Are there existing deep learning models performing this task?
- Data imbalance is an issue for both Virus and DeepSEA datasets. Why were different evaluation metrics chosen for the classification tasks on the two datasets?
- It is unclear how the training for the baseline models like CPC was performed? Was sufficient hyperparameter tuning performed for all the models?
- I am not entirely sure how the model can perform transfer learning from bacteria to virus datasets. Was the model trained using bacteria sequence (or a part of it) re-trained on viral genomes?
- The ablation analysis needs to be more comprehensive. It is also unclear if the variable sequence length was explored thoroughly. It seems that only 2 settings were tried - 150 and 72/78.

Minor points:
- The result section could be arranged better for making it easier to follow
- Typo in Table 6 caption: "din"
- Table 5 results are repeated in Table 8
- Are the ablation study differences for forward and reverse-compliment strands significant?
- The related work section for genomic sequence predictions could include more recent works than DeepSEA and DanQ
- The input length of 150 bp might be limiting the model's performance as studies have shown that longer DNA sequences can better capture the neighboring context of the signal being predicted.


**Summary Of The Paper:**

This paper presents a self-supervised learning approach using contrastive loss for representation learning of genomic sequences. The contrastive loss has been used for self-supervised learning in the NLP and computer vision domains and the paper presents its application for genomics. Self-supervised contrastive learning tries to maximize the agreement between augmented views of a sample. Therefore, Self-GenomeNet splits a sequence into two subsequences, and the learned representation of subsequence 1 is compared to subsequence 2 and its revers compliment (positive samples) as well as other sequences (negative samples). The described method has two considerations specific to the genomics tasks - (1) it handles variable sequences (2) it incorporates reverse complement information of the sequences. The method is applied on two prediction tasks and one transfer learning task using sequences from viral, bacterial, and human genomes. Its performance is compared to the supervised model, generative language model, and self-supervised learning models - CPC and Contrastive-sc. The results show improved classification performance over the baseline for both supervised retraining and semi-supervised training settings.

**Summary Of The Review:**

Overall the ideas presented in the paper are interesting and potentially useful. However, limited experiments and results fail to support the main claims of the paper. The results could include tasks with label scarcity and some of the choices for experimental settings are unclear.

---

> ### Author Response · Authors · 2021-11-19
> **Response to Reviewer ifj3 (1/3)**
>
>
> We thank the reviewer for the useful comments and suggested improvements.
> We improved the writing of the manuscript, included more recent studies, added a new dataset with label scarcity, updated our achieved performance with F1-score.
>
> ### Choice of Dataset:
> > * The paper's central claim is that self-learning-based methods can help with genomics tasks with limited data. However, the datasets used in the paper do not reflect those tasks and have an ample amount of labeled information.
>
> We disagree with the claim that we have not shown performance with data with scarce label information, since artificially restricting the amount of labeled samples, as we have done, is common practice for self-supervised learning tasks. We performed semi-supervised learning experiments with subsets of labeled data (1%, 10%) on two different datasets and showed the ability of learning Self-GenomeNet with limited labeled data. We draw your attention to Table 2 in section 5. Moreover, we performed ablation studies and did several experiments on two different datasets with only 0.1% of the labeled and observed Self-GenomeNet outperformed the supervised model with 10.8% better performance.
>
> However, we agree that another dataset with the actual scarcity of labels is interesting. We have added a **new** dataset for predicting **T6SS effector protein** where the dataset has *naturally scarce labels*, described in Section 4 "Datasets and Tasks". Here is our achieved performance in terms of F1 score and Macro-averaged recall:
>
> | Base Network   | Fine-tuned    | Fine-tuned | Fixed    | Fixed |
> |  --------- | ---------| ----------- | -------  |  ----------- |
> | Method   | Recall-M    | F1 | Recall-M    | F1 |
> | Supervised  |  58.2  |  0.624  | -  |  - |
> | CPC    |  81.8   | 0.855   | 72.3  | 0.725  |
> | Language Model  | 79.1  | 0.825   | 72.5   | 0.752  |
> | Contrastive-sc   | 74.3  | 0.775  | 62.0  | 0.627  |
> | Self-GenomeNet   | **83.4**  | **0.857**  | **79.3**  | **0.796** |
>
>
>
> > * The rationale for the choice of the datasets, models, and baselines is not entirely clear. For example, there are more recent methods (like Basenji) that perform genome sequence classification. So why was DanQ chosen as the model of choice?
> ### Regarding the rationale of the other datasets:
> The *DeepSEA* dataset was chosen since it is a bio-informatic open benchmark dataset that many other deep learning models were evaluated before. Considering the number of labels and size of the dataset we see it as a kind of "ImageNet dataset of bioinformatics".
>
> We chose the *Virus* classification task because it differs from the DeepSEA task in an important way, as it is trying to differentiate an aspect of an entire given genome sequence, not just regions within a sequence. It is also a relevant task from metagenomics, where it is sometimes necessary to distinguish species of short (~150 nt) sequence reads (see below). We have extended the explanation for our choice of datasets in the manuscript in Section 4, *Datasets and Tasks*.
>
> We chose the *Bacteria* dataset for the **transfer learning task**. Bacteria are much better described and studied than their viral counterparts, and the majority of available sequences belong to bacteria which is the reason why viruses are often termed as “viral dark matter” (Krishnamurthy and Wang 2017). Further, bacteria and viruses are not clearly evolutionarily separated, for example, viruses that attack bacteria (bacteriophages) are frequently incorporated in the bacterial chromosome.  The motivation for this transfer learning regime was to make use of the abundance of bacterial data to optimize neural model development for tasks on viruses (such as viral identification or taxonomic classifications).
>
> We extended the **Datasets and Tasks** part of Section 4, describing our motivation for choosing the datasets.

---

> > ### Author Response · Authors · 2021-11-19
> > **Response to Reviewer ifj3 (2/3)**
> >
> > ### 150 bp Input Length
> > > * The input length of 150 bp might be limiting the model's performance as studies have shown that longer DNA sequences can better capture the neighboring context of the signal being predicted.
> >
> > The motivation to limit the input length to 150 nt in some of our evaluations is to be close to a potential application, since this is a typical read length that gets produced during *Next Generation Sequencing* which is frequently used (Quail et al. 2012). The length of 150 nt, therefore, allows a direct application of the sequencing output to perform model predictions that can be built using pre-trained models as presented in our work. We also evaluate some experiments with a sequence length of 1,000 nt, because this is a meaningful length of the output of genome assemblers, where these 150 nt long reads are assembled together using overlap information or mapping to reference sequences to produce longer fragments.
> >
> > The short sequence length can be justified to build neural networks directly on the read output of a sequencer (150 nt), rather than on the output of the assembling process since the latter comes with several limitations. it requires (i) computational resources and often (ii) the presence of *reference genomes*, which are often absent when an environmental sample is sequenced; furthermore (iii) assembling Softwares perform poorly on repetitive regions of bacterial genomes that are of particular interest to study since these often belong to the bacterial defense system. Therefore, traditional bioinformatics software is taking raw read input rather than preprocessed contigs, for example for identification of specific loci of interest such as CRISPR sequences (Skennerton, Imelfort, and Tyson 2013), or for functional profiling to reconstruct metabolic functions (Franzosa et al. 2018).
> >
> > We agree that we should mention the motivation for our values chosen in the manuscript. We added this to Section 8.
> >
> > ### Choice of Model Architecture and Baseline
> > > * The rationale for the choice of the datasets, models, and baselines is not entirely clear. For example, there are more recent methods (like Basenji) that perform genome sequence classification. So why was DanQ chosen as the model of choice?
> > > * The related work section for genomic sequence predictions could include more recent works than DeepSEA and DanQ
> >
> > We thank you for your suggestions on additional references for related literature on this part. We have modified the manuscript and added the corresponding references (Basenji) to our related work section (Section 2).  We also included other recent approaches there, such as iDeepS, DeepBind, DeeperBind, ECBLSTM (Trabelsi et al., 2019) .
> >
> > We have chosen to base the self-GenomeNet architecture on the DanQ approach since a recent benchmark done by Trabelsi et al. (Trabelsi et al., 2019) showed the network composed of CNN, RNN, and embedding layers can achieve the best performance on several different genomic tasks and datasets (which aligns with our experience). A specific feature of architectures involving RNNs that we make use of is that it naturally creates representations of subsequences of different lengths (since the sequential output of a unidirectional RNN incorporates information of subsequences of growing length). Using models such as Basenji, which are based on dilated convolutions, would not give us this possibility. We have modified the manuscript in Section 2, *Deep learning methods for genome sequences* to stress this point more.
> >
> > ### Evaluation Metric
> > > * Data imbalance is an issue for both Virus and DeepSEA datasets. Why were different evaluation metrics chosen for the classification tasks on the two datasets?
> >
> > We agree in retrospect that the choice of different evaluation metrics for the different architectures may appear arbitrary. We chose ROC AUC and PR AUC for DeepSEA dataset in order to be in line with the publications on this dataset. DeepSEA (Zhou, J. et al.), DanQ (Quang D. et al.) and Basenji(Kelley et al.) papers all report ROC AUC, whereas DanQ paper (Quang D. et al.) argues PR AUC metric is less prone to inflation by the class imbalance. For the virus dataset, we had chosen macro average recall as it is easy to interpret, since this metric is equivalent to class-balanced accuracy. However, we now additionally report the F1 metric for experiments on the virus dataset besides macro-averaged recall.
> >
> > ### Significance of the Ablation Study Results
> > > * Are the ablation study differences for forward and reverse-complementary strands significant?
> >
> > We have recently performed new experiments for Forward, Reverse, and Reverse-complement strands. For each, we pretrained the models in self-supervised fashion three times and then trained the pretrained models in supervised way under linear evaluation protocol.
> >
> > | Forward (F1)  |  Reverse(F1)  |  RC (F1) |
> > |  --------- | ---------| ----------- |
> > |  0.7048$\pm$0.0025  | 0.6952$\pm$0.0003  |  0.7089$\pm$ 0.0008 |

---

> > > ### Author Response · Authors · 2021-11-19
> > > **Response to Reviewer ifj3 (3/3)**
> > >
> > > ### Other Weaknesses
> > > > * It is unclear how the training for the baseline models like CPC was performed? Was sufficient hyperparameter tuning performed for all the models?
> > >
> > > We optimized the learning rate for our model as well as for the baseline models. For other hyperparameter choices, we considered the recommendations made in the other baseline models. We modified and added more details about the hyperparameter tuning and optimization of other self-supervised models in*Section 8, Optimization of Self-Supervised Methods* and in*Section 4, Optimization*.
> > >
> > >
> > > > * I am not entirely sure how the model can perform transfer learning from bacteria to virus datasets. Was the model trained using bacteria sequence (or a part of it) re-trained on viral genomes?
> > >
> > > We pre-trained the Self-GenomeNet model on the *Bacteria*-dataset (using contrastive loss) and then went on to use the resulting weights to perform classification on the *Virus* / *DeepSEA* datasets -- either by fixing the pre-trained weights and fitting a linear layer on top or by fine-tuning the models on the classification datasets. We realize that this was not well described in the manuscript, which we have updated at Setion 5 *Transfer to other tasks*.
> > >
> > > > * The ablation analysis needs to be more comprehensive. It is also unclear if the variable sequence length was explored thoroughly. It seems that only 2 settings were tried - 150 and 72/78.
> > >
> > > The point of this step in the ablation study is to determine whether having a model that uses variable sequence lengths has an advantage over a model that does not do that, a binary decision. We see the model that splits the sequence close to the mid-point (but also at a place divisible by 6, the stride length of the first CNN) as the natural choice for a model that does *not* incorporate *variable* sequence lengths. Comparisons between different sequence length splits would, in our opinion, exceed the scope of the ablation study.
> > >
> > > ### Other minor issues
> > > We thank you for pointing us towards some of the minor issues. We have used this opportunity to clean up the manuscript, fix the mentioned and other typos etc.
> > >
> > > ### References:
> > > * Trabelsi, A., Chaabane, M., & Ben-Hur, A. (2019). Comprehensive evaluation of deep learning architectures for prediction of DNA/RNA sequence binding specificities. Bioinformatics, 35(14), i269-i277. DOI 10.1093/bioinformatics/btz339
> > >
> > > * Franzosa, Eric A., Lauren J. McIver, Gholamali Rahnavard, Luke R. Thompson, Melanie Schirmer, George Weingart, Karen Schwarzberg Lipson, et al. 2018. “Species-Level Functional Profiling of Metagenomes and Metatranscriptomes.” Nature Methods 15 (11): 962–68.
> > >
> > > * Krishnamurthy, Siddharth R., and David Wang. 2017. “Origins and Challenges of Viral Dark Matter.” Virus Research 239 (July): 136–42.
> > >
> > > * Quail, Michael A., Miriam Smith, Paul Coupland, Thomas D. Otto, Simon R. Harris, Thomas R. Connor, Anna Bertoni, Harold P. Swerdlow, and Yong Gu. 2012. “A Tale of Three next Generation Sequencing Platforms: Comparison of Ion Torrent, Pacific Biosciences and Illumina MiSeq Sequencers.” BMC Genomics 13 (July): 341.
> > >
> > > * Skennerton, Connor T., Michael Imelfort, and Gene W. Tyson. 2013. “Crass: Identification and Reconstruction of CRISPR from Unassembled Metagenomic Data.” Nucleic Acids Research 41 (10): e105.
> > > *Zhou, J., Troyanskaya, O. Predicting effects of noncoding variants with deep learning–based sequence model. Nat Methods 12, 931–934 (2015).
> > > *David R Kelley, Yakir A Reshef, Maxwell Bileschi, David Belanger, Cory Y McLean, and Jasper Snoek.  Sequential regulatory activity prediction across chromosomes with convolutional neural networks. Genome research , 28(5):739–750, 2018

---

> > > > ### Comment · Reviewer_ifj3 · 2021-11-28
> > > > **Response to the rebuttal**
> > > >
> > > > Thank you for the detailed responses to my comments. I appreciate the additional results on real-world datasets with label scarcity and the clarifications and have revised my review to raise the original score.

---

### Decision · Program_Chairs · 2022-01-20

**Decision:**

Reject

**Comment:**

Based on the contrastive learning loss wildly used in the NLP and computer vision domains, this paper presents Self-GenomeNet, a contrastive learning method for representation learning of genomic sequences.
As shown in the experiment section, the improvement compared to baselines CPC, Language model, and even supervised learning method is considerable, on three benchmark datasets in both self-supervised and semi-supervised evaluation.

Even after the discussion phase, there exists disagreement among the reviewers.
AC considered all reviews, author responses, and the discussions, as well as read the paper.
While the paper has some merit such as an effective Self-GenomeNet model for the particular problem setup, reviewers still have several reservations to directly accepting it:
+ Questionable impact. The proposed framework is overall a simple combination of existing methods and beyond genome datasets, the impact of this proposed method is questionable.
+ Limited inspiration. The proposed method is mainly constructed on the previously-proposed contrastive learning loss wildly-used in the NLP and computer vision domains, the benefits of the proposed method may be limited on the genome data (especially the domain-specific data augmentation e.g., reverse complement). How can the insights foster future research?
+ Lack of justification. The innovations introduced by the paper seem ad-hoc, and the reasons for the large observed improvement are not entirely intuitive.
Meanwhile, even with the provided response from the authors, the connection between motivation and the proposed method is still not crystal clear.

Given the above reservations, AC could not accept the paper for now but encourage the authors to fully revise the paper and strengthen their work.